# Optimal Private Median Estimation
# under Minimal Distributional Assumptions

**Christos Tzamos**
Department of Computer Science
University of Wisconsin-Madison
tzamos@wisc.edu

**Emmanouil V. Vlatakis-Gkaragkounis**
Department of Computer Science
Columbia University
emvlatakis@cs.columbia.edu

**Ilias Zadik**
Center for Data Science
New York University
zadik@nyu.edu

## Abstract

We study the fundamental task of estimating the median of an underlying distribution from a finite number of samples, under pure differential privacy constraints. We focus on distributions satisfying the minimal assumption that they have a positive density at a small neighborhood around the median. In particular, the distribution is allowed to output unbounded values and is not required to have finite moments. We compute the exact, up-to-constant terms, statistical rate of estimation for the median by providing nearly-tight upper and lower bounds. Furthermore, we design a polynomial-time differentially private algorithm which provably achieves the optimal performance. At a technical level, our results leverage a Lipschitz Extension Lemma which allows us to design and analyze differentially private algorithms solely on appropriately defined "typical" instances of the samples.

## 1 Introduction

We consider the problem of estimating the median of a distribution under privacy constraints. Given sample access to an unknown distribution $\mathcal{D}$ with median $m(\mathcal{D})$, the goal is to produce an estimate $\hat{m}$ that satisfies $|m(\mathcal{D}) - \hat{m}| \leq \alpha$ with probability $1 - \beta$ while respecting the privacy of the samples.

Median estimation is a fundamental task in many applications and it is commonly preferred to other location estimation methods, like the mean, due to its robustness properties [Hub81, HRRS11] and high break-down point [DG07, HR, Alo03, TP16]. An additional property of the median is that it can be efficiently estimated using few samples even under heavy tailed distributions whose mean may not even be well-defined. A minimal statistical assumption for estimation is a mild concentration of the distribution around the median, i.e. that for some parameters $L$ and $r$, the distribution has density at least $L$ in an interval $[m(\mathcal{D}) - r, m(\mathcal{D}) + r]$ around the median. In such a case it can be shown that $O(\log(1/\beta)/(\alpha^2 L^2))$ samples suffice to learn the median within $\alpha < r$ with probability $1 - \beta$ [BA20].

Beyond its statistical guarantees an additional important property of an estimator is to be private. In medical applications for instance, samples may correspond to human subjects and it is crucial that the estimator maintains their privacy [DE13, ZLZ+17, FLJ+14, MEO13, YKÖ17] while still producing accurate results.

Differential privacy [Dwo06, DMNS06, DR14] is the leading measure for quantifying privacy guarantees for an arbitrary randomized function of many input points. It is parameterized by a value $\varepsilon$ and requires that changing the value of any input point may change the probability of any output by at most a factor of $e^{\varepsilon}$. More formally we use the Hamming distance $d_H(X, X') := |\{i \in [n] | X_i \neq X_i'\}|$, defined for two $n$-tuples of samples $X, X' \in \mathbb{R}^n$.

**Definition 1.1.** *A randomized algorithm $\mathcal{A}$ is $\varepsilon$-differential private if for all subsets $S \in \mathcal{F}$ of the output space $(\Omega, \mathcal{F})$ and $n$-tuples of samples $X_1, X_2 \in \mathbb{R}^n$, it holds*

$$\mathbb{P}(\mathcal{A}(X_1) \in S) \leq e^{\varepsilon d_H(X_1, X_2))}\mathbb{P}(\mathcal{A}(X_2) \in S). \tag{1.1}$$

While there has been a lot of work in statistical estimation under privacy constraints, obtaining private estimators with optimal sample complexity was only recently shown to be possible. Karwa and Vadhan [KV18] studied the basic problem of learning the parameters of a univariate Gaussian and obtained the optimal rate for estimation. Subsequently, sample efficient estimators were obtained for learning multivariate Gaussians and product distributions [KSSU19, ZKKW20, KLSU19]. Further work obtained minimax rates for various other statistical problems [CKS20, KSU20, DWJ16], including mean estimation for heavy tailed distributions [KSU20, DWJ16], parameter estimation for Erdős-Rényi graphs and stochastic block models [BCS15, BCSZ18b, SU19] and general hypothesis selection [BKSW19, GKK+20].

## 1.1 Our contributions

Our work considers the setting of the recent work by Avella-Medina and Brunel [BA19, BA20] who studied the problem of median estimation under arbitrary distributions $\mathcal{D}$ that satisfy a mild concentration condition and whose median lies in a bounded range.

**Assumption 1.2.** *The distribution $\mathcal{D}$ has a unique median $m(\mathcal{D}) \in [-R, R]$. Moreover, there exist positive constants $r, L$ such that it admits a density with respect to the Lebesgue measure $f(u)$ when $u \in [m(\mathcal{D}) - r, m(\mathcal{D}) + r]$ and furthermore it holds $f(u) \geq L$, for all $u \in [m(\mathcal{D}) - r, m(\mathcal{D}) + r]$.*

Note that under the Assumption 1.2 the distribution can output arbitrarily large values and is not required to have finite moments.

We call distributions that satisfy this Assumption 1.2 *admissible*. For the class of admissible distributions, [BA19] obtained two private estimators that efficiently produce an estimate $\hat{m}$ for the median such that $|m(\mathcal{D}) - \hat{m}| \leq \alpha$ with probability $1 - \beta$ using a sample size $n$. The estimators achieve sample complexity

$$\Omega\left(\frac{\log\frac{1}{\beta}}{L^2\alpha^2} + \frac{\log^2\frac{1}{\beta}\log\frac{1}{\delta}}{\varepsilon^2 L\alpha} + \frac{\log(\frac{R}{\alpha}+1)\log\frac{1}{\delta}}{\varepsilon Lr}\right) \quad \text{and} \quad \Omega\left(\frac{\log\frac{1}{\beta}}{L^2\alpha^2} + \frac{\log^2\frac{1}{\beta}(\log\frac{1}{\beta}+\log\frac{1}{\delta})}{\varepsilon^2 L\alpha}\right)$$

They both work under the $(\varepsilon, \delta)$-differential privacy setting (see Definition A.1) which is a weaker guarantee than pure differential privacy.

Our main result is a significantly improved estimator that learns the median under pure $\varepsilon$-differential privacy.

**Theorem 1.3.** *There exists an $\varepsilon$-differentially private algorithm that draws $n = O\left(\frac{\log(\frac{1}{\beta})}{L^2\alpha^2} + \frac{\log(\frac{1}{\beta})}{\varepsilon L\alpha} + \frac{\log(\frac{R}{\alpha}+1)}{\varepsilon Lr}\right)$ samples from any admissible distribution $\mathcal{D}$ and in $\mathrm{poly}(n)$ time produces an estimate $\hat{m}$ that satisfies $|m(\mathcal{D}) - \hat{m}| \leq \alpha$ with probability $1 - \beta$.*

Our result improves upon several aspects of the prior bound and gives stronger privacy guarantees. It has linear dependence in both $1/\varepsilon$ and $\log(1/\beta)$ as opposed to quadratic/cubic. Importantly, we show that the sample complexity we obtain is tight in all possible parameters up to absolute constants. We obtain matching lower-bounds showing that any estimator that produces an $\alpha$-accurate estimate of the median must use at least a constant fraction of the samples used by our estimator.

**Theorem 1.4.** *Consider any $\varepsilon$-differentially private algorithm. There exists an admissible distribution $\mathcal{D}$ from which $n = \Omega\left(\frac{\log(\frac{1}{\beta})}{L^2\alpha^2} + \frac{\log(\frac{1}{\beta})}{\varepsilon L\alpha} + \frac{\log(\frac{R}{\alpha}+1)}{\varepsilon Lr}\right)$ samples are required to produce an estimate $\hat{m}$ that satisfies $|m(\mathcal{D}) - \hat{m}| \leq \alpha$ with probability $1 - \beta$.*

Our results can also be applied for location estimation in the case of a Gaussian distribution $N(\mu, \sigma^2)$ with known variance $\sigma^2$. In this case, the median is equal to the mean and we recover *the exact tight bounds* obtained in recent work of [KLSU19, KV18]. Thus, despite our minimal distributional assumptions, the trade-off between samples, accuracy and privacy that we obtain is the same as the simpler parametric setting of Gaussian distributions.

## 1.2 Our approach and techniques

To obtain our main result we devise a more general principled framework that can be applied to many other problems in private statistical estimation. Our framework splits the design problem in two steps, and is inspired by earlier work on learning Erdős-Rényi graphs [BCSZ18a, BCSZ18b]:

1. We obtain private algorithms for a subset of the domain. We focus our attention to a restricted class of instances that are "typical" with respect to the distribution. We define a set of constraints that samples should satisfy with good probability and focus only on instances that meet those constraints. In this restricted setting, we obtain an estimator that is accurate and private for these instances only.

2. We extend our estimator to all instances and guarantee privacy globally. We want to do this while preserving the output of our estimator in the set of typical instances.

In the **first step**, we make careful use of the well-studied Laplace mechanism, that is a common method for a turning a non-private algorithm to a private one. It works by adding Laplace noise to the deterministic final output of the algorithm in order to ensure privacy. The amount of noise that needs to be added crucially depends on the sensitivity of the algorithm to changes in the input.

For the task of computing the median of $n$ points, the sensitivity can be quite high as even if all points are bounded in $[-R, R]$ there are instances where changing a single input may change the median by $R$[1].

While a worst case instance might have large sensitivity as argued above, one expects that in a "typical instance" drawn randomly from a concentrated distribution the median does not change so drastically when few of the points move. Indeed, we quantify the notion of a typical instance as the family of instances that have many points in various distances from the sample median and show that even changing a small fraction of input points cannot move the median of the sample dramatically.

To obtain our main result we exploit this observation and initially focus only on typical instances of the distribution. For those instances, we show that a variant of the Laplace mechanism, which we call "flattened" Laplace mechanism ensures that differential privacy is guaranteed when focusing only on typical instances. Focusing on typical instances, suffices for obtaining high probability guarantees but ideally we want that our estimator to ensure privacy even under worst case instances.

In the **second step**, we aim to define the output distribution of our estimator in the atypical instances to ensure privacy globally. To achieve this, we employ a method for Lipschitz extensions that has been developed in prior work for estimating parameters of Erdős-Rényi graphs, called the "Extension Lemma" (Check Proposition 2.1, [BCSZ18a]) The "Extension Lemma" shows that it is always possible to extend a private algorithm from a smaller domain $A$ to a larger domain $B$ without changing the output distribution for instances in $A$. It achieves this by explicitly defining what the output distribution should be for instances in $B \setminus A$ and only worsens the privacy guarantee by a factor of 2. This allows to obtain worst-case private estimators that are guaranteed to produce accurate estimates with high probability over the instances drawn from the distribution.

Finally, an important technical contribution is showing that the extension can be computed in polynomial time. We do this by characterizing the structure of the resulting extended output distributions showing that they are piece-wise constant or exponential. We can identify all these pieces through a simple greedy algorithm and sample exactly from the resulting distribution. This provides the first instance of a natural problem for which the extension given by the Extension Lemma [BCSZ18a] can be computed in polynomial time thus answering affirmatively a question posed by the authors.

### 1.3 Further Related Work

Lipschitz-extensions is a popular technique for designing private algorithms that was introduced in [BBDS13, KNRS13]. A common theme in those methods is that they develop a Lipschitz estimator for a small domain that is extended to be Lipschitz throughout the whole domain and can then be made private through the Laplace mechanism. Developing a Lipschitz median estimator for our setting would lead to suboptimal rates that scale linearly with the range of possible values $R^2$. In contrast, our framework constructs directly a private mechanism for the small domain and extends it to a private mechanism in the whole domain via the more general Extension Lemma that has been developed in recent work [BCSZ18a, BCSZ18b].

The framework of Lipschitz extensions was also recently used for estimation of the median along with other statistics such as the variance and the trimmed mean [CD20]. Their paper focuses on arbitrary data-sets without statistical assumptions. Applied to our setting, their methods would yield sub-optimal sample complexity, scaling linearly with the range possible values $R$, again as opposed to the optimal logarithmic dependence we obtain in our work. Median estimation has also been studied in [DL09], who gave an $(\varepsilon, \delta)$-differentially private mechanism that is consistent in the limit but did not provide explicit non-asymptotic rates.

## 2 Preliminaries: The Extension Lemma

In this Section for the reader's convenience, we present briefly a main tool behind of our approach, which we refer to from now on as the Extension Lemma. The Extension Lemma is proven in [BCSZ18b, BCSZ18a].

Using the notation of Definition 1.1 let us consider an arbitrary $\varepsilon$-differentially private algorithm defined on input belonging in some set $\mathcal{H} \subset \mathbb{R}^n$. Then the Extension Lemma guarantees that the algorithm *can be always extended* to a $2\varepsilon$-differentially private algorithm defined for arbitrary input data from $\mathcal{M}$ with the property that if the input data belongs in $\mathcal{H}$, the distribution of output values is the same with the original algorithm. The result in [BCSZ18a] is generic, in the sense of applying to any input metric space, but here we present it for simplicity only when the input space if the input space is $\mathbb{R}^n$ and is equipped with the Hamming distance $d_H$. Formally the result is as following,

**Proposition 2.1** ("The Extension Lemma" Proposition 2.1, [BCSZ18a]). *Let $\hat{\mathcal{A}}$ be an $\varepsilon$-differentially private algorithm designed for input from $\mathcal{H} \subseteq \mathbb{R}^n$ with arbitrary output measure space $(\Omega, \mathcal{F})$. Then there exists a randomized algorithm $\mathcal{A}$ defined on the whole input space $\mathbb{R}^n$ with the same output space which is $2\varepsilon$-differentially private and satisfies that for every $X \in \mathcal{H}$, $\mathcal{A}(X) \overset{d}{=} \hat{\mathcal{A}}(X)$.*

## 3 A Rate-Optimal Estimator

In this section, we present an optimal $\varepsilon$-differentially private algorithm for median estimation. We defer the proofs of the stated results to the supplementary material. We start with the model of estimation.

### 3.1 The Model of Estimation.

In our statistical model, for some $n \in \mathbb{N}$ one is given $n$ independent identically distributed (i.i.d) samples $X_1, \cdots, X_n$ from a distribution $\mathcal{D}$. We make the assumption that the distribution $\mathcal{D}$ is *admissible* per Assumption 1.2.

We are interested in estimating the median of $\mathcal{D}$, denoted by $m(\mathcal{D})$, from the $n$ samples using an $\varepsilon$-differentially private algorithm. Specifically we consider the following minimax rate:

$$\mathcal{R}(n, \varepsilon)(\alpha) := \min_{\mathcal{A} \text{ is } \varepsilon-\text{D.P.}} \max_{\mathcal{D} \text{ admissible}} \mathbb{P}_{x_1, \ldots, x_n \sim^{\text{i.i.d}} \mathcal{D}} \left[ |\mathcal{A}(x_1, x_2, \ldots, x_n) - m(\mathcal{D})| \geq \alpha \right], \quad (3.1)$$

where in the above $\min$ we use the abbreviation D.P. for differential privacy. Our focus is on *the sample complexity* defined for any $\alpha \in (0, r), \beta \in (0, 1)$ as following.

$$n_{\text{sc}}(\alpha, \beta) := \inf\{n \in \mathbb{N} : \mathcal{R}(n, \varepsilon)(\alpha) \leq \beta\}. \quad (3.2)$$

In words, we want to understand the minimum number of samples that are required for an $\varepsilon$-differentially private estimator to estimate the median with accuracy $\alpha$ with probability $1 - \beta$.

## 3.2 The Optimal Sample Complexity

We offer a tight, up to absolute constants, characterization of the sample complexity. Our results follow by proposing and analyzing the performance of an $\varepsilon$-differentially private algorithm and also proving the corresponding minimax lower bound.

**Theorem 3.1.** *Assume* $\varepsilon \in (0, 1)$. *Then for any* $\alpha \in (0, \min\{R, r\})$, $\beta \in (0, \frac{1}{2})$ *it holds*

$$
n_{\mathrm{sc}}(\alpha, \beta) = \Theta \left( \frac{\log(\frac{1}{\beta})}{L^2 \alpha^2} + \frac{\log\left(\frac{1}{\beta}\right)}{\varepsilon L \alpha} + \frac{\log\left(\frac{R}{\alpha} + 1\right)}{\varepsilon L r} \right).
$$

## 3.3 The optimal algorithm

We describe here the construction of the optimal $\varepsilon$-differentially private algorithm. We use the Extension Lemma and follow the path described in items 1,2 in Section 1.2. Our high-level approach is to first define a a "typical" subset of the input space and a restricted differentially private algorithm defined only on the "typical" subset, and then use the Extension Lemma to extend it to a private algorithm on the whole input space.

**The typical set** We first define a "typical" subset $\mathcal{H} \subseteq \mathbb{R}^n$ of the input space. Let $C > 1$ a constant we are going to choose sufficiently large.

$$
\mathcal{H} = \mathcal{H}_C = \left\{ X \in \mathbb{R}^n : \begin{cases} \sum_{i \in [n]} \mathbf{1}\{X_i - m(X) \in [0, \frac{\kappa C}{Ln}]\} \geq \kappa + 1 \\ \sum_{i \in [n]} \mathbf{1}\{m(X) - X_i \in [0, \frac{\kappa C}{Ln}]\} \geq \kappa + 1 \\ \kappa \in \{1, \cdots, \frac{Lnr}{2C}\} \\ m(X) \in [-R - r/2, R + r/2] \end{cases} \right\}, \tag{3.3}
$$

where by $m(X)$ we denote the left empirical median of the set $(X_1, \cdots, X_n)$ (see Definition A.2). In words, the first constraints assume that there are sufficiently many samples falling sufficiently close to the empirical median $m(X)$, at distances which are multiples of $C/Ln$. The motivation for this choice comes from the fact that each interval $[0, \kappa C/Ln]$ is assigned from an admissible $\mathcal{D}$ at least $C\kappa/n$ probability mass based on the Assumption 1.2. Hence, in expectation, it contains at least $C\kappa > \kappa$ out of the $n$ samples. The last constraint of $\mathcal{H}$ assumes that the $m(X)$ falls sufficiently close to the interval the population median $m(\mathcal{D})$ is assumed to belong to, based on the Assumption 1.2.

**The restricted algorithm** We then define the algorithm on inputs from $\mathcal{H}$ as a randomised algorithm with density given by

$$
f_{\hat{A}(X)}(\omega) = \frac{1}{\hat{Z}} \exp\left( -\frac{\varepsilon}{4} \min\left\{ \frac{Ln}{3C} |m(X) - \omega|, Lrn \right\} \right), \omega \in \mathcal{I} := [-R - 4Cr, R + 4Cr] \tag{3.4}
$$

where the normalizing constant is

$$
\hat{Z} = \int_{\mathcal{I}} \exp\left( -\frac{\varepsilon}{4} \min\left\{ \frac{Ln}{3C} |m(X) - \omega|, Lrn \right\} \right) \mathrm{d}\omega. \tag{3.5}
$$

We call this distribution a "flattened" Laplacian mechanism (see Figure 1). Note that the normalizing constant $\hat{Z}$ does not need to be indexed by the $n$-tuple $X$ as it can be easily proven that for $C > 1/2$ the integral on the left hand side of 3.5 gives the same value for all $X \in \mathcal{H}$. Furthermore, observe that to implement the restricted algorithm given as input a data-set $X \in \mathcal{H}$, we first compute the left empirical median $m(X)$ and then sample from the "flattened" Laplacian distribution, one part of it is a uniform distibution and the other a truncated Laplacian distribution. The exact sampling details from the distribution are defered to Section 4.

The motivation to select this distribution is to ensure that the algorithm is $\varepsilon/2$-differentially private. The reason is that the function $m(X)$ as a function from $(\mathcal{H}, d_X)$ to the reals, is highly sensitive (e.g. it can move distance $\Omega(R)$ with $n$ changes) and therefore adding Laplace noise, as customary in the design of private algorithms, would not suffice. Nevertheless we show that it satisfies the following "approximate" Lipschitz constraint with a significantly improved Lipschitz constant.

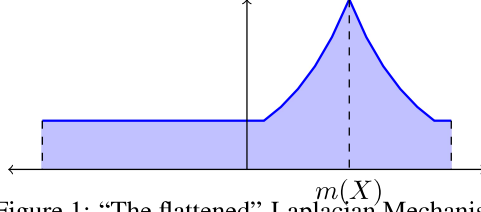

$$m(X)$$

Figure 1: "The flattened"-Laplacian Mechanism

**Lemma 3.2.** *Suppose $X, Y \in \mathcal{H}$ with Hamming distance $d_H(X, Y) \leq Lnr$. Then*

$$|m(X) - m(Y)| \leq \frac{3C}{Ln} d_H(X, Y).$$

Using this constraint and the definition of the "flattened" Laplacian mechanism, it can be shown using elementary arguments that indeed for inputs $X, Y \in \mathcal{H}$ it indeed holds for all real $q$, $f_{\hat{\mathcal{A}}(X)}(q) \leq e^{\frac{\varepsilon}{2} d_H(X,Y)} f_{\hat{\mathcal{A}}(Y)}(q)$, which certifies the $\varepsilon/2$-differential privacy. The following holds.

**Lemma 3.3.** *The algorithm $\hat{\mathcal{A}}$ defined on $\mathcal{H}$ is $\frac{\varepsilon}{2}$-differentially private.*

With respect to accuracy guarantees the following elementary result can be proven by using the density of the "flattened" Laplace distribution, as defined in Equation (3.4).

**Lemma 3.4.** *Suppose $C > 1$ and a fixed $X \in \mathcal{H}$. Then for any $\alpha \in (0, r)$, $\beta \in (0, 1)$ for some $n = O\left( \frac{\log\left(\frac{1}{\beta}\right)}{\varepsilon L \alpha} + \frac{\log\left(\frac{R}{\alpha} + 1\right)}{\varepsilon L r} \right)$ then*

$$\mathbb{P}[|\hat{\mathcal{A}}(X) - m(X)| \geq \alpha] \leq \beta,$$

*where the probability is with respect to the randomness of the algorithm $\hat{\mathcal{A}}$.*

**The general algorithm**   Now we construct the final algorithm using the Extension Lemma. First, note that the algorithm $\hat{\mathcal{A}}$ defined on $\mathcal{H}$ based on Lemma 3.3 is $\frac{\varepsilon}{2}$-differentially private. Hence, using the Extension Lemma there is an $\varepsilon$-differentially private algorithm $\mathcal{A}$ defined on the whole input space $\mathbb{R}^n$ with the property that for any $X \in \mathcal{H}$ it holds $\mathcal{A}(X) \overset{d}{=} \hat{\mathcal{A}}(X)$. This is the algorithm we consider for the upper bound.

An investigation on the proof of the Extension Lemma in [BCSZ18a], deferred due to space constraints to the supplementary material, shows that the algorithm $\mathcal{A}$ can be described as receiving input $X \in \mathbb{R}^n$ and outputting a sample from the continuous distribution with density given by

$$f_{\mathcal{A}(X)}(\omega) = \frac{1}{Z_X} \exp\left( \inf_{X' \in \mathcal{H}} \left[ \frac{\varepsilon}{2} d_H(X, X') - \frac{\varepsilon}{4} \min\left\{ \frac{Ln}{3C} |m(X') - \omega|, Lrn \right\} \right] \right), \quad (3.6)$$

where $\omega \in [-R - 4Cr, R + 4Cr]$ and $Z_X$ is the appropriate normalizing constant.

We now briefly elaborate on its accuracy guarantee. The set $\mathcal{H}$, besides crucial for Lemma 3.2, is important to our construction because of the following technical "typical" guarantee it has.

**Lemma 3.5.** *Let $\mathcal{D}$ be an admissible distribution, $\beta \in (0, 1)$, $n \geq 3$ and $X = (X_1, \ldots, X_n)$ consisting of i.i.d. samples from $\mathcal{D}$. Suppose $C > 5$ satisfies $4Ce \exp\left(-\frac{2C}{27}\right) < 1/2$. Then for some $C' = C'(C) > 0$ and $n = O\left( \frac{\log \frac{1}{\beta}}{L^2 r^2} \right)$ it holds*

$$\mathbb{P}\left( \exists X' \in \mathcal{H} \text{ s.t. } d_H(X, X') \leq C' \log \frac{1}{\beta}, m(X') = m(X) \right) \geq 1 - \beta.$$

In words, the lemma claims that with probability $1 - \beta$ an $n$-tuple of i.i.d. samples from $\mathcal{D}$, call it $X$, has $O\left( \log \frac{1}{\beta} \right)$ Hamming distance from at least one element $X' \in \mathcal{H}$ with $m(X) = m(X')$.

Now the accuracy argument follows from the following high-level idea. By the definition of $\varepsilon$-differential privacy (Definition 1.1) we know that the algorithm $\mathcal{A}$ when applied to two inputs of

sufficiently "small" Hamming distance the outputs are "close" in distribution. Hence, Lemma 3.5 implies that the law of $\mathcal{A}(X)$ with input $X$ is "close" to the law of $\mathcal{A}(X')$ where the input is some $X' \in \mathcal{H}$ with $m(X') = m(X)$. Now since $X' \in \mathcal{H}$ by construction the law of $\mathcal{A}(X')$ equals the law of $\hat{\mathcal{A}}(X')$. Furthermore from Lemma 3.3 we know that $\hat{\mathcal{A}}(X')$, and therefore also $\mathcal{A}(X)$, concentrates around $m(X') = m(X)$. Finally standard concentration results (see e.g. [BA20, Lemma 3]) imply that with $n = \Omega\left(\log \frac{1}{\beta}/(L^2\alpha^2)\right)$ samples it holds $|m(X) - m(\mathcal{D})| \leq \alpha$ with probability $1 - \beta$.

Using the above idea we obtain the following result.

**Theorem 3.6.** *Suppose $C > 0$ is sufficiently large, $\varepsilon \in (0,1)$, $\mathcal{D}$ is an admissible distribution and $\mathcal{A}$ is the $\varepsilon$-differentially private algorithm defined above. Then for any $\alpha \in (0, r)$ and $\beta \in (0, 1)$ for some*

$$n = O\left(\frac{\log\left(\frac{1}{\beta}\right)}{L^2\alpha^2} + \frac{\log\left(\frac{1}{\beta}\right)}{\varepsilon L \alpha} + \frac{\log\left(\frac{R}{\alpha} + 1\right)}{\varepsilon L r}\right)$$

*it holds* $\quad \mathbb{P}_{X_1, X_2, \ldots, X_n \overset{iid}{\sim} \mathcal{D}}[|\mathcal{A}(X_1, \ldots, X_n) - m(\mathcal{D})| \geq \alpha] \leq \beta.$

Finally, the discussion on the computationally efficient implementation of the algorithm $\mathcal{A}$ is deferred to Section 4.

### 3.4 Lower Bounds

In this subsection we discuss the lower bound part on $n_{\text{sc}}(\alpha, \beta)$ as stated in Theorem 3.1. We establish that all three terms are necessary for all values of the parameters, using a different method for each one of them. We omit the complete proofs for the supplementary material and provide only brief hints for the methods followed. First, we start with the $\varepsilon$-independent part (the "non-private" part).

**Proposition 3.7.** *Let $L > 0, \varepsilon \in (0,1), R, r > 0$ with $Lr \leq \frac{1}{2}$ and $\alpha \in (0, \min\{r, R\}), \beta \in (0, \frac{1}{2})$. Suppose $n = o\left(\log\left(\frac{1}{\beta}\right)/(L^2\alpha^2)\right)$. Then for any algorithm $\mathcal{A}$ there exists an admissible distribution $\mathcal{D}$ for this specific value of the parameters with the property*

$$\mathbb{P}_{X_1, X_2, \ldots, X_n \overset{iid}{\sim} \mathcal{D}}[|\mathcal{A}(X) - m(\mathcal{D})| \geq \alpha] > \beta.$$

The proposition follows by showing that learning a a Bernoulli random variable at accuracy $\gamma > 0$ with probability $1 - \beta$ reduces to learning the median of an admissible distribution at accuracy $\gamma L$ with probability $1 - \beta$. Then using that the sample complexity of learning a Bernoulli distribution is known to require $\Theta\left(\log\left(\frac{1}{\beta}\right)/\gamma^2\right)$ samples we conclude the proof.

We now state the lower bound on the first term on the $\varepsilon$-dependent part.

**Proposition 3.8.** *Let $L > 0, \varepsilon \in (0,1), R, r > 0$ with $Lr \leq \frac{1}{2}$ and $\alpha \in (0, \min\{r, \frac{R}{2}\}), \beta \in (0, \frac{1}{2})$. Suppose $n = o\left(\log\left(\frac{1}{\beta}\right)/(\varepsilon L \alpha)\right)$. Then for any $\varepsilon$-differentially private algorithm $\mathcal{A}$ there exists an admissible distribution $\mathcal{D}$ for this specific value of the parameters with the property*

$$\mathbb{P}_{X_1, X_2, \ldots, X_n \overset{iid}{\sim} \mathcal{D}}[|\mathcal{A}(X) - m(\mathcal{D})| \geq \alpha] > \beta.$$

The proof of this lower bound follows by a "local" argument. We construct two admissible distribution $\mathcal{D}, \mathcal{D}'$ with medians of distance $O(\alpha)$ which we can couple so that drawing $X, X'$ two i.i.d. $n$-tuples from $\mathcal{D}, \mathcal{D}'$ respectively it holds $d_H(X, X') = O(\alpha L n)$ with high probability. Using the definition of differential privacy we conclude that $\mathcal{A}(X), \mathcal{A}(X')$ assign the same probability to each event up to a multiplicative factor $e^{O(\varepsilon \alpha L n)}$. Yet, we prove that an algorithm contradicting the assumptions of Proposition 3.8 will imply that while $\mathcal{A}(X)$ assigns mass at least $1 - \beta$ on the interval of width $2\alpha$ around $m(\mathcal{D})$, the algorithm $\mathcal{A}(X')$ needs to assign mass at most $\beta$. Hence combining the above implies $e^{-\varepsilon \alpha L n} = O(\beta)$ or $n = \Omega\left(\log\left(\frac{1}{\beta}\right)/(\varepsilon L \alpha)\right)$.

**Proposition 3.9.** *Let $L > 0, \varepsilon \in (0,1), R, r > 0$ with $Lr \leq \frac{1}{2}$ and $\alpha \in (0, \min\{r, R\}), \beta \in (0, \frac{1}{2})$. Suppose $n = o\left(\log\left(\frac{R}{\alpha} + 1\right)/(\varepsilon L r)\right)$. Then for any $\varepsilon$-differentially private algorithm $\mathcal{A}$ there exists an admissible distribution $\mathcal{D}$ for this specific value of the parameters with the property*

$$\mathbb{P}_{X_1, X_2, \ldots, X_n \overset{iid}{\sim} \mathcal{D}}[|\mathcal{A}(X) - m(\mathcal{D})| \geq \alpha] > \beta.$$

This lower bound follows by a "global" argument. We show that any $\varepsilon$-differentially private algorithm contradicting the conclusion of Proposition 3.9 needs to assign at least $\Omega\left(e^{-Lrn}\right)$ probability mass in at least $\frac{R}{\alpha}+1$ distinct intervals in $[-R, R]$. Since the total probability mass must sum up to one we obtain the desired lower bound.

## 4 A $\mathrm{poly}(n)$-time implementation of the optimal algorithm

Having established the existence of an optimal-rate differential private estimator, in this section we present in high-level a way that it can be efficiently implemented in polynomial time. For concision, we defer the precise statement and proofs to the paper's supplement.

First, notice that in the case of a data-set $X$ which would belong *a-priori* to the typical set $\mathcal{H}$, our estimator corresponds to sampling from the "flattened-Laplacian" continuous distribution of $\hat{\mathcal{A}}(X')$ as defined in Equation (3.4). Therefore, its implementation would correspond to a simple two-phase protocol. In the first round we would flip an appropriately biased coin to decide between the two regions, the region where the estimator samples proportional to the Laplacian density and the region where the estimator samples proportional to the uniform distribution. In the second round we would simply apply conditional sampling either from a Laplacian or a uniform distribution, depending on the outcome of the coin flip.

However, the final algorithm which is defined not only on typical data-sets $X \in \mathcal{H}$, employs the Extension Lemma (Proposition 2.1) which introduces an extra computational challenge. More precisely, using Equation (3.6), implementing efficiently the extended mechanism $\mathcal{A}$ requires $\mathrm{poly}(n)$-time sampler [3] for the (appropriately normalized) distribution:

$$\mathrm{UNNORMALIZED}(X, \omega) = \exp\left(\inf_{X' \in \mathcal{H}}\left[\left(\frac{\varepsilon}{2}d_H(X, X')\right) - \frac{\varepsilon}{4}\min\left\{\frac{Ln}{3C}\left|m(X') - \omega\right|, Lrn\right\}\right]\right)$$

defined in $[-B, B] = [-R - 4Cr, R + 4Cr]$.

Our main technical contribution for this part is to show that the above underlying constrained optimization problem can be solved in polynomial-time and then show how to use it to obtain a polynomial-time sampler for the desired distribution. A key observation is that $\mathrm{UNNORMALIZED}(X, \omega)$ depends only on the possible values of the median $m(X')$ and the distance $d_H(X, X')$, for any data-set $X' \in \mathcal{H}$. To simplify our notation, we introduce the notion of the $\mathrm{TYPICALHAMMING}$ distance of a data-set $X$ and the typical set $\mathcal{H}$:

**Lemma 4.1.** *Let* $\mathrm{TYPICALHAMMING}(X, \xi) = \min_{X' \in \mathcal{H}, m(X') = \xi} d_H(X, X')$. *Then for any data-set $X$,* $\mathrm{TYPICALHAMMING}(X, \cdot)$ *is piece-wise constant function in* $[-R - r/2, R + r/2]$ *with at most* $\mathrm{poly}(n)$ *changes. Additionally, for any $\xi$ and data-set $X$,* $\mathrm{TYPICALHAMMING}(X, \xi)$ *can be computed exactly using a* $\mathrm{poly}(n)$-*time algorithm.*

Leveraging the partition of the interval $[-B, B]$ implied by the different constant parts of Lemma 4.1 we are able to show the following for $\mathrm{UNNORMALIZED}(X, \omega)$:

**Lemma 4.2.** *For any given data-set $X$, there is a partition of $[-B, B]$ to a collection $\mathbb{J}$ of $\mathrm{poly}(n)$ consecutive intervals,* $\mathbb{J} = \{J_1, \cdots, J_{\mathrm{poly}(n)}\}$ *such that :*

$$\textit{For any } J_i \in \mathbb{J} \quad : \mathrm{UNNORMALIZED}(X, \omega) = \exp\left(\alpha_i \omega + \beta_i\right) \quad \forall \omega \in J_i$$

*Moreover, we can compute exactly $\mathbb{J}$ and the constants $(\alpha_i, \beta_i)$ for every $J_i \in \mathbb{J}$ in $\mathrm{poly}(n)$ time.*

In words, Lemma 4.2 implies that our final mechanism consists a concatenation of multiple ($\mathrm{poly}(n)$) piecewise different exponentials and constants.

Therefore, its implementation would correspond to a simple two-phase protocol. In the first round we would sample from a discrete distribution on a $\mathrm{poly}(n)$-cardinality domain to decide among the regions $J_1, \cdots, J_{\mathrm{poly}(n)}$. In the second round we would simply apply conditional sampling from the corresponding either exponential or uniform which will be truncated on the region which was the outcome of the first round.

# 5 The Broader Impact of Our Work

Aggregating large amounts of data has been important for quantitative social scientists. However increasing amounts of data also increase privacy risks. *Do handy medical apps put in jeopardy my insurance policy or ability to get a mortgage? Can I be harmed by my census responses? If I agreed to having committed a traffic violation in a survey then I would promptly receive a fine and a criminal record?*

The invasion of this area of questioning is nowadays ubiquitous, obligating even the governments to adopt new and sweeping privacy laws such as the EU's General Data Protection Regulation 2016/679 (GDPR); However, even if extra confidentiality pledges have been added in research with statements that information will be released only at a group level and not at an individual one , the majority of the participants of those studies doubt their validity. Additionally, privacy scientists have proposed different types of tracing attacks indicating that releasing the summary statistics can result in as much as breach as publishing the private micro data. For example, it could be possible that if many tables are produced for a specific data-set, then an appropriate combination of the tables could leak information about one person with a specific profile.

To mitigate these risks, many researchers had proposed a solution in the concept of differential privacy. In this framework researchers reported the outcome of their analyses having firstly injected a well-calibrated noise. More precisely, the level of the noise has been chosen in such a way that it would be computationally arduous to reconstruct any personal micro data but simultaneously would permit an accurate estimation of group statistics.

**In our work**, we present an optimal differential private estimator with an efficient implementation for *median estimation*, one of the most preferred statistical index in political and medical sciences. Despite its introduction as a term is relatively recent (the earliest trace in English literature appears to be in 1881 by Francis Galton in one of his surveys), it seems that as a measure of central tendency it behaves more robustly than many other indices such as mean and mode. For example, the nominal median income reflects much better the real-life setting than the mean corresponding index. Indeed calculating the cut-off where half of the households earn more, and half earn less is much more robust and illustrative than the mean estimator which could easily have been biased by a small number of billionaires. In psychology where the majority of data is inherently categorical, lacking in real continuum spectrum, the median estimator consists the proper way of summarizing different case studies. Concerning even the task of the determination of the critical high-risk age groups during a pandemic, median estimator plays a significant role in balancing out some possible outliers.

In all these cases, median gives crucial information propagating probably several social, political and financial implications. More importantly, however, safeguarding the individual privacy of the members of a survey could restrict acts of unfair discrimination based on the leaked personal data.

# 6 Acknowledgements

Christos Tzamos would like to acknowledge the support of NSF grants CCF-2008006. Emmanouil Vasileios Vlatakis Gkaragkounis was supported by NSF grants CCF-1703925, CCF-1763970, CCF-1814873, CCF-1563155, and by the Simons Collaboration on Algorithms and Geometry and Ilias Zadik was supported by CDS-Moore-Sloan Postdoctoral Fellowship. We are grateful to Adam Smith for bringing this problem to our attention. E.V would like to thank Lampros Flokas, Ana-Andrea Stoica and Myrto Pantelaki for the helpful discussions at camera-ready stage of this project. This work was also supported in part by the Onassis Foundation - Scholarship ID: F ZN 010-1/2017-2018.

## Footnotes

[1]For example if $\lceil n/2 \rceil$ points are located at $-R$ and $\lfloor n/2 \rfloor$ points are located in $R$ changing a single point can move the median from $-R$ to $R$.

[2]This is because the Lipschitz-constant must be at least $R/n$ as changing all the input should produce any possible median.

[3] In terms of computational complexity, we follow the convention that arithmetic operations execute in $O(1)$ time.

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
