[Supplementary Material]

# Contents

# A  Basic Definitions and the Extension Lemma

## A.1  Basic Definitions

A similar notion to $\varepsilon$-differential privacy (Definition 1.1) is the notion of $(\varepsilon, \delta)$-differential privacy.

**Definition A.1.** *A randomized algorithm $\mathcal{A}$ is $(\varepsilon, \delta)$-differential private if for all subsets $S \in \mathcal{F}$ of the output measurable space $(\Omega, \mathcal{F})$ and n-tuples of samples $X_1, X_2 \in \mathbb{R}^n$ it holds*

$$\mathbb{P}\left(\mathcal{A}(X_1) \in S\right) \leq e^{\varepsilon d_H(X_1, X_2)} \mathbb{P}\left(\mathcal{A}(X_2) \in S\right) + \delta. \tag{A.1}$$

It is rather straightforward that $(\varepsilon, \delta)$-differential privacy is a *weaker* notion to $\varepsilon$-differential privacy, in the sense that for any $\varepsilon, \delta > 0$ any $\varepsilon$-differentially private algorithm is also an $(\varepsilon, \delta)$-differentially private algorithm.

Of importance to us in the design of the algorithm desicred in the main body of this work is the notion of *the left (empirical) median* of an $n$-tuple of samples.

**Definition A.2.** *For $x = (x_1, \ldots, x_n) \in \mathbb{R}^n$, we denote by $x_{(1)}, \ldots, x_{(n)}$ the reordered coordinates of $x$ in nondecreasing order, i.e. $\min_{1 \leq i \leq n} x_i = x_{(1)} \leq \ldots \leq x_{(n)} = \max_{1 \leq i \leq n} x_i$. We let $\ell = \lfloor n/2 \rfloor$ and $m(x) = x_{(l)}$ be the empirical (left) median of $x$.*

## A.2  On the Extension Lemma

In this section we provide more details on the Extension Lemma as stated in Proposition 2.1. We repeated it for convenience.

**Proposition A.3** ("The Extension Lemma" Proposition 2.1, [BCSZ18a]). *Let $\hat{\mathcal{A}}$ be an $\varepsilon$-differentially private algorithm designed for input from $\mathcal{H} \subseteq \mathbb{R}^n$ with arbitrary output measure space $(\Omega, \mathcal{F})$. Then there exists a randomized algorithm $\mathcal{A}$ defined on the whole input space $\mathbb{R}^n$ with the same output space which is $2\varepsilon$-differentially private and satisfies that for every $X \in \mathcal{H}$, $\mathcal{A}(X) \overset{d}{=} \hat{\mathcal{A}}(X)$.*

Albeit the generality of the result, as mentioned in the conclusion of [BCSZ18a], the Extension Lemma does not provide any guarantee that the extension of the algorithm $\hat{\mathcal{A}}$ can be made in a computationally efficient way. In this work, as described in the main body of the paper, we show that the Extension Lemma is applicable in the context of median estimation, and furthermore the extension can be implemented in polynomial time. Hence, naturally, to perform this extension in a computationally efficient way we don't use the Extension Lemma as a "blackbox" but rather need to use the specific structure of the extended private algorithm, by digging into the proof of it in [BCSZ18a, Proposition 2.1.].

**The density of the extended algorithm - general**   For simplicity, we present the density of the extended algorithm under certain assumptions that will be correct in the context of our work. As everywhere in this paper, let us first focus on the case the input space is $\mathcal{M} = \mathbb{R}^n$ equipped with the Hamming distance and the output space is the real line equipped with the Lebesgue measure. Furthermore let us assume also that for any $X \in \mathcal{H}$ the randomised restricted algorithm $\hat{\mathcal{A}}(X)$ follows a real-valued continuous distribution with a density $f_{\hat{\mathcal{A}}(X)}$ with respect to the Lebesque measure, given by Equation (3.4). We repeat here the definitions for convenience;

$$f_{\hat{\mathcal{A}}(X)}(\omega) = \frac{1}{\hat{Z}} \exp\left(-\frac{\varepsilon}{4} \min\left\{\frac{Ln}{3C}|m(X) - \omega|, Lrn\right\}\right), \omega \in \mathcal{I} := [-R - 4Cr, R + 4Cr] \tag{A.2}$$

where the normalizing constant is

$$\hat{Z} = \int_{\mathcal{I}} \exp\left(-\frac{\varepsilon}{4} \min\left\{\frac{Ln}{3C}|m(X) - \omega|, Lrn\right\}\right) d\omega. \tag{A.3}$$

As claimed in Section 3.3 the normalizing constant $\hat{Z}$ does not have an index $X$ because the right hand side of (A.3) takes the same value for any $X \in \mathcal{H}$. This follows by the following Lemma.

**Lemma A.4.** *Suppose $C > 1/2$. Then for any $X \in \mathcal{H}$,*

$$\int_{\mathcal{I}} \exp\left(-\frac{\varepsilon}{4} \min\left\{\frac{Ln}{3C} |m(X) - \omega|, Lrn\right\}\right) d\omega = \int_{\mathcal{I}} \exp\left(-\frac{\varepsilon}{4} \min\left\{\frac{Ln}{3C} |\omega|, Lrn\right\}\right) d\omega.$$

Then from the proof of the Extension Lemma [Section 4, [BCSZ18a]] we have that the "extended" $\varepsilon$-differentially private algorithm $\mathcal{A}$ on input $X \in \mathbb{R}^n$ admits also a density given by

$$f_{\mathcal{A}(X)}(\omega) = \frac{1}{Z_X} \inf_{X' \in \mathcal{H}} \left[\exp\left(\frac{\varepsilon}{4} d_H(X, X')\right) f_{\hat{\mathcal{A}}(X')}(\omega)\right], \omega \in \mathbb{R} \tag{A.4}$$

where

$$Z_X := \int_{\mathbb{R}} \inf_{X' \in \mathcal{H}} \left[\left(\frac{\varepsilon}{4} d_H(X, X')\right) f_{\hat{\mathcal{A}}(X)'}(\omega)\right] d\omega.$$

For reasons of completeness we state the corollary of the Extension Lemma that establishes that in our setting the algorithm $\mathcal{A}$ satisfies the desired properties of the Extension Lemma.

**Proposition A.5.** *Under the above assumptions, the algorithm $\mathcal{A}$ is $\varepsilon$-differentially private and for every $X' \in \mathcal{H}$, $\mathcal{A}(X') \stackrel{d}{=} \hat{\mathcal{A}}(X')$.*

We also include the proof of the Proposition in the following section, which essentially follows the proof of the Extension Lemma [BCSZ18a, Proposition 2.1.] adapted to our case.

As a technical remark note that for the density in equation (A.4) to be well-defined we require that for every $X \in \mathbb{R}^n$ the "unnormalized" density function

$$G_X(\omega) := \inf_{X' \in \mathcal{H}} \left[\exp\left(\frac{\varepsilon}{2} d_H(X, X')\right) f_{\hat{\mathcal{A}}(X')}(\omega)\right], \omega \in \mathbb{R} \tag{A.5}$$

is integrable and has a finite integral as well. Both conditions follows by the following Lemma, which establishes - among other properties - that $G_X$ is a continuous function almost everywhere and has a finite integral.

**Lemma A.6.** *Suppose the above assumptions hold and fix $X \in \mathbb{R}^n$. Then,*

- *$G_X(\omega) = 0$ for all $\omega \notin \mathcal{I} = [-R - 4Cr, R + 4Cr]$*

- *$G_X$ is $\mathcal{R}$-Lipschitz on $\mathcal{I}$ with Lipschitz constant $\mathcal{R} = e^{\frac{\varepsilon n}{2}} \frac{\varepsilon Ln}{12C\hat{Z}}$, where $\hat{Z}$ is given in (A.3).*

*Furthermore, it holds $0 \leq \int_{\omega \in \mathbb{R}} G_X(\omega) d\omega \leq 1$.*

Plugging in now the densities of the restricted algorithm (Equations (A.2)) to the general extended algorithm (Equations (A.6)) and finally using also that according to Lemma A.4 the normalizing constant for the restricted algoriths is independent of $X' \in \mathcal{H}$, we have

$$f_{\mathcal{A}(X)}(\omega) = \frac{1}{Z_X} \exp\left(\inf_{X' \in \mathcal{H}} \left[\frac{\varepsilon}{2} d_H(X, X') - \frac{\varepsilon}{4} \min\left\{\frac{Ln}{3C} |m(X') - \omega|, Lrn\right\}\right]\right), \tag{A.6}$$

where $\omega \in [-R - 4Cr, R + 4Cr]$ and $Z_X$ is the appropriate normalizing constant.

## A.3 Proof of Proposition A.5

We start with using the [BCSZ18a, Lemma 4.1.] applied in our setting, which gives the following Lemma.

**Lemma A.7.** *Let $\mathcal{A}'$ be a real-valued randomized algorithm designed for input from $\mathcal{H}' \subseteq \mathbb{R}^n$. Suppose that for any $X \in \mathcal{H}'$, $\mathcal{A}'(X)$ admits a density function with respect to the Lebesque measure $f_{\mathcal{A}'(X)}$. Then the following are equivalent*

*(1) $\mathcal{A}'$ is $\varepsilon$-differentially private on $\mathcal{H}$;*

*(2) For any $X, X' \in \mathcal{H}$*

$$f_{\mathcal{A}'(X)}(\omega) \leq e^{\varepsilon d_H(X,X')} f_{\mathcal{A}'(X')}(\omega), \tag{A.7}$$

*almost surely with respect to the Lebesque measure.*

We now proceed with the proof of Proposition A.5.

*Proof of Proposition A.5.* We first prove that $\mathcal{A}$ is $\varepsilon$-differentially private over all pairs of input from $\mathbb{R}^n$. Using Lemma A.7 it suffices to prove that for any $X_1, X_2 \in \mathcal{H}$,

$$f_{\mathcal{A}(X_1)}(\omega) \leq \exp\left(\varepsilon d_H(X_1, X_2)\right) f_{\mathcal{A}(X_2)}(\omega),$$

almost surely with respect to the Lebesque measure. We establish it in particular for every $\omega \in \mathbb{R}$. Notice that if $\omega \notin \mathcal{I}$, both sides are zero from Lemma A.6. Hence let us assume $\omega \in \mathcal{I}$. Let $X_1, X_2 \in \mathbb{R}^n$. Using triangle inequality we obtain for every $\omega \in \mathcal{I}$,

$$\inf_{X' \in \mathcal{H}} \left[ \exp\left(\frac{\varepsilon}{2} d_H(X_1, X')\right) f_{\hat{\mathcal{A}}(X')}(\omega) \right] \leq \inf_{X' \in \mathcal{H}} \left[ \exp\left(\frac{\varepsilon}{2} \left[d_H(X_1, X_2) + d_H(X_2, X')\right]\right) f_{\hat{\mathcal{A}}(X')}(\omega) \right]$$

$$= \exp\left(\frac{\varepsilon}{2} d_H(X_1, X_2)\right) \inf_{X' \in \mathcal{H}} \left[ \exp\left(\frac{\varepsilon}{2} d_H(X, X')\right) f_{\hat{\mathcal{A}}(X')}(\omega) \right],$$

which implies that for any $X_1, X_2 \in \mathcal{M}$,

$$Z_{X_1} = \int_\Omega \inf_{X' \in \mathcal{H}} \left[ \exp\left(\frac{\varepsilon}{2} d(X_1, X')\right) f_{\hat{\mathcal{A}}(X')}(\omega) \right] d\omega$$

$$\leq \exp\left(\frac{\varepsilon}{2} d(X_1, X_2)\right) \int_\Omega \inf_{X' \in \mathcal{H}} \left[ \exp\left(\frac{\varepsilon}{2} d(X_2, X')\right) f_{\hat{\mathcal{A}}(X')}(\omega) \right] d\omega$$

$$= \exp\left(\frac{\varepsilon}{2} d(X_1, X_2)\right) Z_{X_2}.$$

Therefore using the above two inequalities we obtain that for any $X_1, X_2 \in \mathbb{R}^n$ and $\omega \in \mathcal{I}$,

$$f_{\mathcal{A}(X_1)}(\omega) = \frac{1}{Z_{X_1}} \inf_{X' \in \mathcal{H}} \left[ \exp\left(\frac{\varepsilon}{2} d_H(X_1, X')\right) f_{\hat{\mathcal{A}}(X')}(\omega) \right]$$

$$\leq \frac{1}{\exp\left(-\frac{\varepsilon}{2} d_H(X_2, X_1)\right) Z_{X_2}} \exp\left(\frac{\varepsilon}{2} d_H(X_1, X_2)\right) \inf_{X' \in \mathcal{H}} \left[ \exp\left(\frac{\varepsilon}{2} d(X_2, X')\right) f_{\hat{\mathcal{A}}(X')}(\omega) \right]$$

$$= \exp\left(\frac{\varepsilon}{2} d(X_1, X_2)\right) \frac{1}{Z_{X_2}} \inf_{X' \in \mathcal{H}} \left[ \exp\left(\frac{\varepsilon}{2} d_H(X_2, X')\right) f_{\hat{\mathcal{A}}(X')}(\omega) \right]$$

$$= \exp\left(\varepsilon d_H(X_1, X_2)\right) f_{\mathcal{A}(X_2)}(\omega),$$

as we wanted.

Now we prove that for every $X \in \mathcal{H}$, $\mathcal{A}(X) \stackrel{d}{=} \hat{\mathcal{A}}(X)$. Consider an arbitrary $X \in \mathcal{H}$. We know that $\hat{\mathcal{A}}$ is $\varepsilon/2$-differentially private which based on Lemma A.7 implies that for any $X, X' \in \mathcal{H}$

$$f_{\hat{\mathcal{A}}(X)}(\omega) \leq \exp\left(\frac{\varepsilon}{2} d_H(X, X')\right) f_{\hat{\mathcal{A}}(X')}(\omega), \tag{A.8}$$

almost surely with respect to the Lebesque measure. Observing that the above inequality holds almost surely as equality if $X' = X$ we obtain that for any $X \in \mathcal{H}$ it holds

$$f_{\hat{\mathcal{A}}(X)}(\omega) = \inf_{X' \in \mathcal{H}} \left[ \exp\left(\frac{\varepsilon}{2} d_H(X, X')\right) f_{\hat{\mathcal{A}}(X')}(\omega) \right],$$

almost surely with respect to the Lebesque measure. Using that $f_{\hat{\mathcal{A}}(X)}$ is a probability density function we conclude that in this case

$$Z_X = \int f_{\hat{\mathcal{A}}(X)}(\omega) d\omega = 1.$$

Therefore

$$f_{\hat{\mathcal{A}}(X)}(\omega) = \frac{1}{Z_X} \inf_{X' \in \mathcal{H}} \left[ \exp\left(\varepsilon d_H(X, X')\right) f_{\hat{\mathcal{A}}(X')}(\omega) \right],$$

almost surely with respect to the Lebesque measure and hence

$$f_{\hat{\mathcal{A}}(X)}(\omega) = f_{\mathcal{A}(X)}(\omega),$$

almost surely with respect to the Lebesque measure. This suffices to conclude that $\hat{\mathcal{A}}(X) \stackrel{d}{=} \mathcal{A}(X)$ as needed.

The proof of Proposition A.5 is complete. $\square$

## A.4 Omitted Proofs

*Proof of Lemma A.4.* Since $X = 0 \in \mathcal{H}$ and the left hand side for $X = 0$ evaluates to the right hand side, it suffices to show that the left hand side does not depend on the value of $m(X)$.

By denoting $\mathcal{I} - m(X) := [-R - 4Cr - m(X), R + 4Cr - m(X)]$ we have

$$\int_{\mathcal{I}} \exp\left(-\frac{\varepsilon}{4} \min\left\{\frac{Ln}{3C}\left|m(X) - \omega\right|, Lrn\right\}\right) \mathrm{d}\omega = \int_{\mathcal{I} - m(X)} \exp\left(-\frac{\varepsilon}{4}\min\left\{\frac{Ln}{3C}\left|\omega\right|, Lrn\right\}\right) \mathrm{d}\omega$$

$$= \int_{(\mathcal{I} - m(X)) \cap \{|\omega| \le 3Cr\}} \exp\left(-\frac{\varepsilon}{4}\frac{Ln}{3C}\left|\omega\right|\right) \mathrm{d}\omega + \int_{(\mathcal{I} - m(X)) \cap \{|\omega| > 3Cr\}} \exp\left(-\frac{\varepsilon}{4}Lrn\right) \mathrm{d}\omega.$$

Now since $X \in \mathcal{H}$ we have $m(X) \in [-R - r/2, R + r/2]$. Hence, using $C > 1/2$ we have $[-3Cr, 3Cr] \subseteq \mathcal{I} - m(X)$. Therefore the last summation of integrals simplifies to

$$\int_{|\omega| \le 3Cr} \exp\left(-\frac{\varepsilon}{4}\frac{Ln}{3C}\left|\omega\right|\right) \mathrm{d}\omega + (|\mathcal{I}| - 6Cr)\exp\left(-\frac{\varepsilon}{4}Lrn\right) \mathrm{d}\omega,$$

which does not depends on $X$ as we wanted. □

*Proof of Lemma A.6.* First notice that if $\omega \notin \mathcal{I}$, from (3.4) for any $X' \in \mathcal{H}$, $f_{\hat{\mathcal{A}}(X')}(\omega) = 0$. Therefore indeed

$$0 \le f_{\mathcal{A}(X)}(\omega) \le \exp\left(\frac{\varepsilon}{2}d_H(X, X')\right) f_{\hat{\mathcal{A}}(X')}(\omega) = 0.$$

We prove now that for all $X' \in \mathcal{H}$, the function $\exp\left(\frac{\varepsilon}{2}d_H(X, X')\right) f_{\hat{\mathcal{A}}(X')}(\omega)$ is $\mathcal{R}$-Lipschitz on $\mathcal{I}$. The claim then follows by the elementary real analysis fact that the pointwise infimum over an arbitrary family of $\mathcal{R}$-Lipschitz functions is an $\mathcal{R}$-Lipschitz function.

Now recall that for all $a, b > 0$ by elementary calculus, $|e^{-a} - e^{-b}| \le |a - b|$. Hence, for fixed $X' \in \mathcal{H}$, using the definition of the density in equation (A.2), we have for any $\omega, \omega' \in \mathcal{I}$,

$$|f_{\hat{\mathcal{A}}(X')}(\omega) - f_{\hat{\mathcal{A}}(X')}(\omega')| \le \frac{\varepsilon}{4\hat{Z}}\left|\min\left\{\frac{Ln}{3C}\left|m(X) - \omega\right|, Lrn\right\} - \min\left\{\frac{Ln}{3C}\left|m(X) - \omega'\right|, Lrn\right\}\right|$$

Now combining with Property B.0.1 we conclude

$$|f_{\hat{\mathcal{A}}(X')}(\omega) - f_{\hat{\mathcal{A}}(X')}(\omega')| \le \frac{\varepsilon}{4\hat{Z}}\left|\min\left\{\frac{Ln}{3C}\left|\omega - \omega'\right|, Lrn\right\}\right| \le \frac{\varepsilon Ln}{12C\hat{Z}}|\omega - \omega'|.$$

In particular, $\exp\left(\frac{\varepsilon}{2}d_H(X, X')\right) f_{\hat{\mathcal{A}}(X')}(\omega)$ is $\mathcal{R} = e^{\frac{\varepsilon n}{2}}\frac{\varepsilon Ln}{12C\hat{Z}}$-Lipschitz since $\exp\left(\frac{\varepsilon}{2}d_H(X, X')\right)$ is a constant independent of $\omega$ with $\exp\left(\frac{\varepsilon}{2}d_H(X, X')\right) \le \exp(\frac{\varepsilon n}{2})$. The proof of the Lipschitz continuity is complete. The final part follows from the fact that $G$ is non-negative by definition and again by definition for arbitrary fixed $X' \in \mathcal{H}$, $f_{\hat{\mathcal{A}}(X')}$ integrates to one and upper bounds pointwise the function $G_X$. □

# B  Proofs for Section 3.3: The Rate-Optimal Algorithm

## B.1  An auxiliary property

The following elementary property is important in what follows, which appeared as Lemma 9.2. in [BCSZ18b].

**Property B.0.1.** *For any $a, b > 0$ the function $f : \mathbb{R} \to \mathbb{R}$, with $f(x) = \min\{a|x|, b\}$, for all $x \in \mathbb{R}$, satisfies the triangle inequality, $f(x + y) \le f(x) + f(y)$ for all $x, y \in \mathbb{R}$.*

## B.2 Proof of Lemma 3.2

*Proof.* We assume $\kappa := d_H(X,Y) \geq 1$ as if it equals zero the Lemma follows.

Consider two data-sets $\begin{cases} X &:= \{x_1, x_2, \ldots, x_\kappa, C_1, \ldots, C_{n-\kappa}\} \\ Y &:= \{y_1, y_2, \ldots, y_\kappa, C_1, \ldots, C_{n-\kappa}\} \end{cases} \in \mathcal{H}$. Without loss of generality we assume that the common part is sorted in an increasing order, i.e. it holds $C_1 \leq \ldots \leq C_{n-\kappa}$.

Notice that the left median of data-set can be altered by the addition of $M'$ new points only to some point among the $\lfloor M'/2 \rfloor + 1$ points on the right and the $\lfloor M'/2 \rfloor + 1$ points on the left of it. Hence the interval $[C_{\lfloor \frac{n-\kappa}{2} \rfloor}, m(X)] \cup [m(X), C_{\lfloor \frac{n-\kappa}{2} \rfloor}]$ (respectively the interval $[C_{\lfloor \frac{n-\kappa}{2} \rfloor}, m(Y)] \cup [m(Y), C_{\lfloor \frac{n-\kappa}{2} \rfloor}]$) there can be at most $\lfloor \frac{\kappa}{2} \rfloor + 1$ points of the data-set $X$ (respectively of the data-set $Y$).

Therefore, leveraging the definition of sensitivity set, we have that

$$|m(X) - C_{\lfloor \frac{n-\kappa}{2} \rfloor}| \leq (\lfloor \frac{\kappa}{2} \rfloor + 1)\frac{C}{Ln}$$

and

$$|m(Y) - C_{\lfloor \frac{n-\kappa}{2} \rfloor}| \leq (\lfloor \frac{\kappa}{2} \rfloor + 1)\frac{C}{Ln},$$

which implies by the triangle inequality

$$|m(X) - m(Y)| \leq (2\lfloor \frac{\kappa}{2} \rfloor + 2)\frac{C}{Ln} \leq 3\kappa\frac{C}{Ln}.$$

$\square$

## B.3 Proof of Lemma 3.3

*Proof.* Using Lemma 3.2 we have for any $X, Y \in \mathcal{H}$,

$$\min\left\{\frac{Ln}{3C}|m(X) - m(Y)|, Lrn\right\} \leq d_H(X,Y). \tag{B.1}$$

We will firstly analyze the non-normalized ratio corresponding to two data-sets $X, Y$:

$$
\begin{aligned}
\frac{Z_X f_{\hat{A}(X)}(q)}{Z_Y f_{\hat{A}(Y)}(q)} &= \frac{\exp\left(\frac{\varepsilon}{4}\min\left\{\frac{Ln}{3C}|m(X) - q|, Lrn\right\}\right)}{\exp\left(-\frac{\varepsilon}{4}\min\left\{\frac{Ln}{3C}|m(Y) - q|, Lrn\right\}\right)} \\
&\leq \exp\left(\frac{\varepsilon}{4}\min\left\{\frac{Ln}{3C}|m(X) - m(Y)|, Lrn\right\}\right) \quad &\text{Property B.0.1} \\
&\leq \exp\left(\frac{\varepsilon}{4}d_H(X,Y)\right) \quad &\text{Equation (B.1)}
\end{aligned}
$$

Furthermore for the ratio of the two normalizing constants we have

$$
\begin{aligned}
\frac{Z_Y}{Z_X} &= \frac{\int_{\mathcal{I}} \exp\left(-\frac{\varepsilon}{4}\min\left\{\frac{Ln}{3C}|m(Y) - q|, Lrn\right\}\right) \mathrm{d}q}{\int_{\mathcal{I}} \exp\left(-\frac{\varepsilon}{4}\min\left\{\frac{Ln}{3C}|m(X) - q|, Lrn\right\}\right) \mathrm{d}q} \\
&\leq \frac{\int_{\mathcal{I}} \exp\left(-\frac{\varepsilon}{4}\min\left\{\frac{Ln}{3C}|m(Y) - q|, Lrn\right\}\right) \mathrm{d}q}{\int_{\mathcal{I}} \exp\left(-\frac{\varepsilon}{4}\left\{\begin{matrix} \min\left\{\frac{Ln}{3C}|m(X) - m(Y)|, Lrn\right\} \\ + \\ \min\left\{\frac{Ln}{3C}|m(Y) - q|, Lrn\right\} \end{matrix}\right\}\right) \mathrm{d}q} \quad &\text{Property B.0.1} \\
&= \exp\left(\frac{\varepsilon}{4}\min\left\{\frac{Ln}{3C}|m(X) - m(Y)|, Lrn\right\}\right) \\
&\leq \exp\left(\frac{\varepsilon}{4}d_H(X,Y)\right) \quad &\text{Equation (B.1)}
\end{aligned}
$$

Combining the two final inequalities above we conclude

$$f_{\hat{\mathcal{A}}(X)}(q) \leq \exp\left(\frac{\varepsilon}{2}d_H(X,Y)\right) f_{\hat{\mathcal{A}}(Y)}(q), \ \forall q \in \mathcal{I}$$

The proof is complete.

$\square$

## B.4 Proof of Lemma 3.4

Lemma 3.4 follows from the following more general lemma where $C > 1$ is allowed to grow with $n$ while in the context of Lemma 3.4 is considered in the asymptotic analysis as a constant.

**Lemma B.1.** *Suppose $C > 1$, possibly scaling with $n$, and $\alpha \in (0, r), \beta \in (0, 1)$. For some*

$$n = O\left(C\frac{\log\left(\frac{1}{\beta}\right)}{\varepsilon L\alpha} + \frac{\log\left(\frac{R}{\alpha}+1\right)}{\varepsilon Lr}\right)$$

*it holds*

$$\max_{X \in \mathcal{H}} \mathbb{P}[|\hat{\mathcal{A}}(X) - m(X)| \geq \alpha] \leq \beta,$$

*where the probability is with respect to the randomness of the algorithm $\hat{\mathcal{A}}(X)$.*

*Proof.*

- **Step 1**: Observe: Since $C > 1/4$ it holds $R + 4Cr > R + r$. Hence, by change of variables $q = q - m(X)$,

$$Z_X = \int_{[-R-4Cr,R+4Cr]} \exp\left(-\frac{\varepsilon}{4}\min\left\{\frac{Ln}{3C}|m(X)-q|, Lrn\right\}\right)\mathrm{d}q$$

$$\geq \int_0^{R+r} \exp(-\frac{\varepsilon}{4}\min\{\frac{Ln}{3C}q, Lrn\})\mathrm{d}q$$

$$= \int_0^{\min\{3rC,R+r\}} \exp(-\frac{\varepsilon}{4}\cdot\frac{Ln}{3C}q)\mathrm{d}q + \int_{\min\{3rC,R+r\}}^{R+r} \exp(-\frac{\varepsilon}{4}Lrn)\mathrm{d}q$$

$$\geq \int_0^{\min\{3rC,R+r\}} \exp(-\frac{\varepsilon}{4}\cdot\frac{Ln}{3C}q)\mathrm{d}q$$

$$= \frac{1}{\frac{\varepsilon}{4}\cdot\frac{Ln}{3C}}\left(1 - \exp(-\frac{\varepsilon}{4}\cdot\frac{Ln}{3C}\min\{3Cr, R+r\})\right)$$

$$= \frac{12C}{n\varepsilon L}\left(1 - \exp(-\frac{n\varepsilon L}{12C}\cdot\min\{3Cr, R+r\}))\right)$$

$$\geq \frac{12C}{n\varepsilon L}\left(1 - \exp(-\Theta\left(\frac{n\varepsilon Lr}{C}\right)))\right)$$

- **Step 2**: Now using that $m(X) \in [-R - r/2, R + r/2]$ and by change of variables $q = q - m(X)$ we have:

$$\mathbb{P}[|\hat{\mathcal{A}}(X) - m(X)| \geq \alpha] \leq \frac{2}{Z_X}\int_\alpha^{2R+3Cr+r} q\exp(-\frac{\varepsilon}{4}\min\{\frac{Ln}{3C}q, Lrn\})\mathrm{d}q$$

$$\leq \frac{2}{Z_X}\left(\int_\alpha^{+\infty} \exp(-\frac{\varepsilon}{4}\frac{Ln}{3C}q)\mathrm{d}q + \int_{3Cr}^{2R+3Cr+r} \exp(-\frac{\varepsilon}{4}Lrn)\mathrm{d}q\right)$$

$$\leq \frac{2}{Z_X}\left(\frac{12C}{n\varepsilon L}\exp(-\frac{\varepsilon}{4}\frac{Ln}{3C}\alpha) + (2R+r)\exp(-\frac{Lr\varepsilon n}{2})\right)$$

Hence we conclude for all $X \in \mathcal{H}$

$$\mathbb{P}[|\hat{\mathcal{A}}(X) - m(X)| \geq \alpha] \leq 2 \left(1 - \exp(-\Theta\left(\frac{n\varepsilon Lr}{C}\right))\right)^{-1} \left(\exp(-\frac{\varepsilon}{4}\frac{Ln}{3C}\alpha) + (2R+r)\frac{n\varepsilon L}{12C}\exp(-\frac{\varepsilon Lrn}{2}))\right)$$

From this and elementary asymptotics we conclude that for $C > 1$ with

$$n = O\left(C\frac{\log\left(\frac{1}{\beta}\right)}{\varepsilon L\alpha} + \frac{C + \log\left(\frac{R}{r} + 1\right)}{\varepsilon Lr}\right)$$

it holds for all $X \in \mathcal{H}$, $\mathbb{P}[|\hat{\mathcal{A}}(X) - m(X)| \geq \alpha] \leq \beta$. Using that $\alpha \leq r$ the above sample complexity bound simplifies to

$$n = O\left(C\frac{\log\left(\frac{1}{\beta}\right)}{\varepsilon L\alpha} + \frac{\log\left(\frac{R}{\alpha} + 1\right)}{\varepsilon Lr}\right).$$

The proof of the Lemma is complete.

$\square$

## B.5   Proof of Lemma 3.5

The following Lemma holds.

**Lemma B.2.** *Let $\mathcal{D}$ be an admissible distribution and $X = (X_1, \ldots, X_n)$ consisting of i.i.d. samples from $\mathcal{D}$. Suppose $C > 5$, possibly scaling with $n$, which satisfies $4Ce\exp\left(-\frac{2C}{27}\right) < 1/2$. Then for any $T \in [0, \frac{Lr}{4C}n]$ it holds*

$$\mathbb{P}\left(\bigcap_{\kappa \in [T, \frac{Lr}{2C}n] \cap \mathbb{Z}} \sum_{i \in [n]} \mathbf{1}\{X_i - m(X) \in [0, \frac{C\kappa}{Ln}]\} \geq \kappa + 1\right) \geq 1 - O\left((8Ce\exp\left(-\frac{C}{8}\right))^{\lceil T \rceil} + e^{-\Theta(L^2 r^2 n)}\right),$$

*and*

$$\mathbb{P}\left(\bigcap_{\kappa \in [T, \frac{Lr}{2C}n] \cap \mathbb{Z}} \sum_{i \in [n]} \mathbf{1}\{X_i - m(X) \in [-\frac{C\kappa}{Ln}, 0]\} \geq \kappa + 1\right) \geq 1 - O\left((8Ce\exp\left(-\frac{C}{8}\right)))^{\lceil T \rceil} + e^{-\Theta(L^2 r^2 n)}\right),$$

We now use Lemma B.2 to prove the following more general version of Lemma 3.5 where $C > 0$ is allowed to potentially scale with $n$.

**Lemma B.3.** *Let $\mathcal{D}$ be an admissible distribution, $\beta \in (0, 1)$, $n \geq 3$ and $X = (X_1, \ldots, X_n)$ consisting of i.i.d. samples from $\mathcal{D}$. Suppose $C > 5$, possibly scaling with $n$, which satisfies $4Ce\exp\left(-\frac{2C}{27}\right) < 1/2$. Then for some $E > 0$, $C' = C'(C) > 0$ with $C'(C) = \Theta(\frac{1}{C})$ as $C$ grows, if $n \geq E\frac{\log\frac{1}{\beta}}{L^2 r^2}$ then it holds*

$$\mathbb{P}\left(\exists X' \in \mathcal{H} \text{ s.t. } d_H(X, X') \leq C'\log\frac{1}{\beta}, m(X') = m(X)\right) \geq 1 - \beta.$$

*Proof of Lemma B.3.* We begin by applying Lemma B.2. First, notice that since $Lr \leq 1/2$ we can take $E > 0$ sufficiently large, so that if $n \geq E\frac{\log\frac{1}{\beta}}{L^2 r^2}$ then $n = \Omega\left(\frac{\log\frac{1}{\beta}}{Lr}\right)$. Hence we can choose $T = \frac{C'(C)}{2}\log\frac{1}{\beta}$ for appropriate $C'(C) > 0$ with $C'(C) = \Theta(1/C)$ as $C$ grows, so that using Lemma B.2 both the two probabilistic guarantees hold with probability at least $1 - \frac{\beta}{2} - O\left(e^{-\Theta(L^2 r^2 n)}\right)$.

Taking now sufficiently large $E > 0$ with $n \geq E \frac{\log \frac{1}{\beta}}{L^2 r^2}$ we can make sure the probabilistic guarantees hold with probability at least $1 - \beta$. Combining the above, with probability at least $1 - \beta$ the following event holds: for all $\kappa \in [T, \frac{Lr}{2C} n] \cap \mathbb{Z}$,

$$\sum_{i \in [n]} \mathbf{1}\{X_i - m(X) \in [-\frac{C\kappa}{Ln}, 0]\} \geq \kappa + 1 \tag{B.2}$$

and

$$\sum_{i \in [n]} \mathbf{1}\{X_i - m(X) \in [-\frac{C\kappa}{Ln}, 0]\} \geq \kappa + 1. \tag{B.3}$$

In words, all but, the closest to $m(X)$, $T = \frac{C'(C)}{2} \log \frac{1}{\beta}$ intervals on the left of $m(X)$, and all but, the closest to $m(X)$, $T = \frac{C'(C)}{2} \log \frac{1}{\beta}$ interval on the right of $m(X)$, satisfy their corresponding constraints described in the typical set $\mathcal{H}$.

Now let us consider a data-set $X$ for which (B.2) and (B.3) hold. We show that given these events one can construct an $X' \in \mathcal{H}$ satisfying the conditions described in the event considered in Lemma B.3. First recall that as $T \leq \frac{Lr}{4C} n$ since $Lr \leq 1/2$ and $C > 5$, it necessarily holds $T \leq n/40$. Since we assume $n \geq 3$ it holds $T < n/2 - 2$. Now, as there are at least $n/2 - 2$ points on the left of $m(X)$ and $n/2 - 2$ points on the right of $m(X)$ we can modify $X$ by choosing arbitrary $T$ points on the right of $m(X)$ and arbitrary $T$ points on the left of $m(X)$ and change all their position to $m(X)$. Notice that such a change produces a new data-set $X'$ which has Hamming distance $2T = C'(C) \log \frac{1}{\beta}$ with $X$ and has the same median with $X$, $m(X) = m(X')$.

Notice that as we moved points closer to the median, all satisfied constraint from $\mathcal{H}$ by the data-set $X$ according to (B.2) and (B.3) continue to be satisfied from $X'$. On top of this, since at least $2T$ points of $X'$ take now exactly the value of the left empirical median, the data-set $X'$ necessarily satisfies also the, potentially violated by $X$, constraints corresponding to the $T$ intervals on the left of $m(X') = m(X)$ and the $T$ intervals on the right of $m(X') = m(X)$. We conclude that $X'$ satisfies all the constraints described in $\mathcal{H}$ and therefore, $X' \in \mathcal{H}$, which completes the proof. $\square$

For the rest part we focus on proving Lemma 3.5.

*Proof of Lemma B.2.* We start by noticing that by identical reasoning as the derivation of inequality (14) of [BA19] in the proof of Lemma 3 in [BA19] we have for $t = r/2 \in [0, r]$

$$\mathbb{P}\left(|m(X) - m(\mathcal{D})| \geq r/2\right) \leq 2e^{-nL^2r^2/8}. \tag{B.4}$$

No we focus on proving the first out of the two probabilistic guarantee,

$$\mathbb{P}\left(\bigcap_{\kappa \in [T, \frac{Lr}{2C} n] \cap \mathbb{Z}} \sum_{i \in [n]} \mathbf{1}\{X_i - m(X) \in [0, \frac{C\kappa}{Ln}]\} \geq \kappa + 1\right) \geq 1 - O\left((8Ce \exp\left(-\frac{C}{8}\right))^{\lceil T \rceil} + e^{-\Theta(L^2 r^2 n)}\right),$$

as the second follows naturally by the symmetric argument around zero. We make the following three observations.

First, observe that generating i.i.d. samples from $\mathcal{D}$ and then determining the left empirical median $m(X)$ based on the realization of $X_1, \ldots, X_n$ is equal in law with first sampling the place of the left empirical median $m(X)$ based on its distribution and sampling uniformly at random an $X_i$ so that $X_i = m(X)$ and then generating $\lceil \frac{n-1}{2} \rceil - 1$ ($n$ is even) i.i.d. samples from $\mathcal{D}$ conditional to be on the left of $m(X)$ and generating $\lceil \frac{n-1}{2} \rceil$ i.i.d. samples from $\mathcal{D}$ conditional to be on the right of $m(X)$. To model the underlying the randomness we can define for each $i$, the trinary random variables where $C_i = -1$ if the sample $i$ is chosen to be on the left of $m(X)$, $C_i = 0$ if the sample equals $m(X)$ and $C_i = 1$ if they are chosen to be on the right.

Second, using the fact that $\mathcal{D}$ has density lower bounded by $L$ in $[m(\mathcal{D}) - r, m(\mathcal{D}) + r]$, observe that our distribution $\mathcal{D}$ can be decomposed as a mixture,

$$\mathcal{D} = \frac{rL}{2} \text{Unif}[m(\mathcal{D}) - r, m(\mathcal{D}) + r] + \left(1 - \frac{rL}{2}\right) \mathcal{D}' \tag{B.5}$$

for some other probability measure $\mathcal{D}'$ on the reals. Now using the first observation, we first sample the position of $m(X)$ and describe the sampling of $X_i$ as follows. If $C_i = 0$ we simply set $X_i = m(X)$. For the other two cases, we assume $C_i = 1$ as the other case is symmetric. If $m(X) - m(\mathcal{D}) \geq r$ then we sample from the distribution $\left(1 - \frac{rL}{2}\right) \mathcal{D}'$ conditional on being on the right of $m(X)$. If $m(X) - m(\mathcal{D}) \leq r$ then we first flip a coin $B_i \overset{d}{=} \text{Bernoulli}(\frac{rL}{2})$ and with probability $\frac{rL}{2}$ (that is when $B_i = 1$) we sample from $\text{Unif}[m(\mathcal{D}) - r, m(\mathcal{D}) + r]$ conditional on being on the right of $m(X)$ and with probability $1 - \frac{rL}{2}$ (that is when $B_i = 0$) we sample from some distribution $\mathcal{D}'$ conditional on being on the right of $m(X)$.

Third, observe that the event of interest becomes less probable when we restrict ourselves to only a subset of the $n$ samples. Hence, since we want to prove a lower bound on the probability of the event we can restrict ourselves to an arbitrary subset.

Now, using (B.4) by neglecting an event of probability $2e^{-nL^2 r^2/8} \leq e^{-\Theta(L^2 r^2 n)}$ from now on we condition on $|m(X) - m| \leq r/2$. Now using the three observations above, we restrict ourselves only on the $n_1 \leq n$ samples that satisfy $B_i = C_i = 1$, that is they are samples from $\text{Unif}[m(\mathcal{D}) - r, m(\mathcal{D}) + r]$ conditional on being on the right of $m(X)$. We denote these samples by $X_1, \ldots, X_{n_1}$ for simplicity. Notice that $\text{Unif}[m(\mathcal{D}) - r, m(\mathcal{D}) + r]$ conditional on being on the right of $m(X)$ is just distributed as $\text{Unif}[m(X), m(\mathcal{D}) + r]$. Hence, conditioning on the value of $n_1$, the density of the conditional distribution of each $X_1, \ldots, X_{n_1}$ given the $m(X)$ can be straightforwardly check to satisfy

$$\frac{2}{3r} \leq f_{X_i}(u|m(X), |m(X) - m(\mathcal{D})| \leq r/2) \leq \frac{2}{r}, u \in [m(X), m(\mathcal{D}) + r]. \qquad \text{(B.6)}$$

Combining the above, to prove our result it suffices to prove

$$\mathbb{P}\left( \bigcap_{\kappa \in [T, \frac{Lr}{2C}n] \cap \mathbb{Z}} \sum_{i \in [n_1]} \mathbf{1}\{X_i - m(X) \in [0, \frac{C\kappa}{Ln}]\} \geq \kappa + 1 \right) \geq 1 - O\left( (8Ce \exp\left(-\frac{C}{8}\right)))^{\lceil T \rceil} + e^{-\Theta(L^2 r^2 n)} \right).$$

Since $m(X) = X_i$ for some $i$ by definition of the left empirical median, it suffices to prove that for

$$J_\kappa := (m(X), m(X) + \frac{C\kappa}{Ln}], \ \kappa = \lceil T \rceil, \lceil T \rceil + 1, \ldots, \lfloor \frac{Lr}{2C}n \rfloor - 1$$

and sequence of events

$$A_\kappa := \{ \sum_{i \in [n_1]} \mathbf{1}\{X_i \in J_\kappa\} \geq \kappa \}, \ \kappa = \lceil T \rceil, \lceil T \rceil + 1, \ldots, \lfloor \frac{Lr}{2C}n \rfloor - 1$$

it holds

$$\mathbb{P}\left( \bigcap_{\kappa = \lceil T \rceil}^{\lfloor \frac{Lr}{2C}n \rfloor - 1} A_\kappa \Big| |m(X) - m(\mathcal{D})| \leq r/2 \right) \geq 1 - O\left( (8Ce \exp\left(-\frac{C}{8}\right)))^{\lceil T \rceil} + e^{-\Theta(L^2 r^2 n)} \right).$$

Using a union bound it suffices to show

$$\sum_{\kappa = \lceil T \rceil}^{\lfloor \frac{Lr}{2C}n \rfloor - 1} \mathbb{P}\left( A_\kappa^c \cap \bigcap_{s = \lceil T \rceil}^{\kappa - 1} A_s \Big| |m(X) - m(\mathcal{D})| \leq r/2 \right) \leq O\left( (8Ce \exp\left(-\frac{C}{8}\right)))^{\lceil T \rceil} + e^{-\Theta(L^2 r^2 n)} \right).$$

Observe that as we are conditioning on $|m(X) - m(\mathcal{D})| \leq r/2$ for all $\kappa$ of interest

$$J_\kappa \subseteq [m(\mathcal{D}) - r, m(\mathcal{D}) + r].$$

Hence using (B.6) we have that, conditioned on $n_1$, for each $i \in [n_1]$ and $\kappa$,

$$\mathbb{P}\left( X_i \in J_\kappa | m(X), |m(X) - m(\mathcal{D})| \leq r/2 \right) \in [\frac{2}{3r}|J_\kappa|, \frac{2}{r}|J_\kappa|]. \qquad \text{(B.7)}$$

Furthermore recall that $n_1 \overset{d}{=} \mathrm{Binom}\left(\lfloor\frac{n-1}{2}\rfloor, \frac{Lr}{2}\right)$ and that it holds $\kappa \leq \frac{Lrn}{2C}$. Since $C > 5$ by standard concentration inequalities, we have that with probability $1 - e^{-\Theta\left(L^2 r^2 n\right)}$, it holds for all $\kappa$ of interest

$$\kappa < \frac{Lrn}{9} < \frac{n_1}{2} < Lrn. \tag{B.8}$$

In what follows we condition on the event (B.8). Suppose $\kappa = \lceil T \rceil$. Then using (B.7) by definition the probability of $A^c_{\lceil T \rceil}$ is at most the probability a sample from a binomial distribution with $N := n_1$ draws and probability $\frac{2}{r}|J_{\lceil T \rceil}| = \frac{2C\lceil T \rceil}{nLr}$ is at most $\lceil T \rceil$. Now conditioning on $n_1$ and denoting the Binomial random variable by $Z$ we have by the additive form of the Chernoff's inequality,

$$\mathbb{P}\left(A^c_{\lceil T \rceil}\middle| m(X), |m(X) - m(\mathcal{D})| \leq r/2, n_1\right) \leq \mathbb{P}\left(Z \leq \lceil T \rceil\right)$$

$$\leq \exp\left(-\frac{\left(\frac{2n_1 C\lceil T \rceil}{Lnr} - \lceil T \rceil\right)^2}{2n_1 \frac{2C\lceil T \rceil}{Lnr}\left(1 - \frac{2C\lceil T \rceil}{Lnr}\right)}\right)$$

Using now that we condition on (B.8) it holds $\frac{2Lnr}{9} < n_1 < 2Lrn$ and $T < \frac{Lnr}{4C}$ we have

$$\mathbb{P}\left(A^c_{\lceil T \rceil}\middle| |m(X) - m(\mathcal{D})| \leq r/2\right) = \mathbb{E}_{m(X), n_1}\left[\mathbb{P}\left(A^c_{\lceil T \rceil}\middle| m(X), |m(X) - m(\mathcal{D})| \leq r/2, n_1\right)\right]$$

$$\leq \exp\left(-\frac{\left(\frac{4}{9}C - 1\right)^2 \lceil T \rceil^2}{2C\lceil T \rceil}\right)$$

which since $C > 5 > \frac{9}{7}$ gives that

$$\mathbb{P}\left(A^c_{\lceil T \rceil}\middle| |m(X) - m(\mathcal{D})| \leq r/2\right) \leq \exp\left(-\frac{C\lceil T \rceil}{8}\right). \tag{B.9}$$

Now for every $\kappa > \lceil T \rceil$ notice that the event $A^c_\kappa \cap \bigcap_{s=0}^{\kappa-1} A_s$ can happen if and only if $\kappa$ of the samples belong in $J_{\kappa-1}$ and the rest $n_1 - \kappa$ samples belong to $J^c_\kappa$. Therefore using the above observations and (B.7),

$$\mathbb{P}\left(A^c_\kappa \cap \bigcap_{s=\lceil T \rceil}^{\kappa-1} A_s\middle| m(X), |m(X) - m(\mathcal{D})| \leq r/2, n_1\right) \leq \binom{n_1}{\kappa}\left(\frac{2}{r}|J_{\kappa-1}|\right)^\kappa \left(1 - \frac{2}{3r}|J_\kappa|\right)^{n_1-\kappa}$$

$$\leq \binom{n_1}{\kappa}\left(\frac{2C(\kappa-1)}{nLr}\right)^\kappa \left(1 - \frac{2C\kappa}{3Lrn}\right)^{n_1-\kappa}$$

$$\leq \left(\frac{n_1 e}{\kappa}\right)^\kappa \left(\frac{2C\kappa}{Lrn}\right)^\kappa \exp\left(-C\frac{2(n_1-\kappa)\kappa}{3Lrn}\right)$$

$$\leq \left(2Ce\frac{n_1}{Lrn}\right)^\kappa \exp\left(-C\frac{2(n_1-\kappa)\kappa}{3Lrn}\right),$$

where we used the elementary inequalities $\binom{m}{m'} \leq \left(\frac{me}{m'}\right)^{m'}$, $1 + x \leq e^x$ and that by definition $n_1 \leq n$.

The last displayed inequality conditioned on the event (B.8) implies

$$\mathbb{P}\left(A_\kappa^c \cap \bigcap_{s=\lceil T \rceil}^{\kappa-1} A_s \,\Big|\, |m(X) - m(\mathcal{D})| \le r/2\right) = \mathbb{E}_{m(X), n_1}\left[\mathbb{P}\left(A_\kappa^c \cap \bigcap_{s=0}^{\kappa-1} A_s \,\Big|\, m(X), |m(X) - m(\mathcal{D})| \le r/2, n_1\right)\right]$$

$$\le \mathbb{E}_{n_1}\left[(4Ce)^\kappa \exp\left(-\frac{2C}{3}\frac{(n_1 - \frac{n_1}{2})\kappa}{Lrn}\right)\right]$$

$$\le \mathbb{E}_{n_1}\left[(4Ce)^\kappa \exp\left(-\frac{2C}{3}\frac{\frac{n_1}{2}\kappa}{Lrn}\right)\right]$$

$$\le \mathbb{E}_{n_1}\left[(4Ce)^\kappa \exp\left(-\frac{2C}{27}\kappa\right)\right]$$

$$= \left(4Ce\exp\left(-\frac{2C}{27}\right)\right)^\kappa \qquad (\text{B.10})$$

Using (B.9) for the first term and since $4Ce\exp\left(-\frac{2C}{27}\right) < 1/2$ a geometric summation over $\kappa \ge \lceil T \rceil$ for the rest terms, we have that conditional on (B.8),

$$\sum_{\kappa = \lceil T \rceil}^{\lfloor \frac{Lr}{2C} n \rfloor - 1} \mathbb{P}\left(A_\kappa^c \cap \bigcap_{s=\lceil T \rceil}^{\kappa-1} A_s \,\Big|\, |m(X) - m(\mathcal{D})| \le r/2\right) \le \exp\left(-\frac{C\lceil T \rceil}{8}\right) + (8Ce\exp\left(-\frac{2C}{27}\right))^{\lceil T \rceil}.$$

Taking now into account the probability of the conditioned event we have (B.8),

$$\sum_{\kappa = \lceil T \rceil}^{\lfloor \frac{Lr}{2C} n \rfloor - 1} \mathbb{P}\left(A_\kappa^c \cap \bigcap_{s=\lceil T \rceil}^{\kappa-1} A_s \,\Big|\, |m(X) - m(\mathcal{D})| \le r/2\right) \le \exp\left(-\frac{C\lceil T \rceil}{8}\right) + (8Ce\exp\left(-\frac{2C}{27}\right))^{\lceil T \rceil} + e^{-\Theta(L^2 r^2 n)}$$

$$\le 2(8Ce\exp\left(-\frac{C}{8}\right))^{\lceil T \rceil} + e^{-\Theta(L^2 r^2 n)}$$

where in the last line since $\frac{C}{8} > \frac{2C}{27}$ and $8Ce > 1$ under our assumptions it holds for all $T \ge 0$

$$\exp\left(-\frac{C\lceil T \rceil}{8}\right) + (8Ce\exp\left(-\frac{2C}{27}\right))^{\lceil T \rceil} \le 2(8Ce\exp\left(-\frac{C}{8}\right))^{\lceil T \rceil}.$$

The proof is complete.

$\square$

## B.6  Proof of Theorem 3.6

We establish instead the following slightly more general result which does not assume that $C$ is a constant, but could scale with $n$.

**Theorem B.4.** *Suppose $C > 0$, possibly scaling with $n$, is bigger than a sufficiently large constant, $\varepsilon \in (0, 1)$, $\mathcal{D}$ is an admissible distribution and $\mathcal{A}$ is the $\varepsilon$-differentially private algorithm defined above. Then for any $\alpha \in (0, r)$ and $\beta \in (0, 1)$ for some*

$$n = O\left(\frac{\log\left(\frac{1}{\beta}\right)}{L^2 \alpha^2} + C\frac{\log\left(\frac{1}{\beta}\right)}{\varepsilon L \alpha} + \frac{\log\left(\frac{R}{\alpha} + 1\right)}{\varepsilon L r}\right)$$

*it holds* $\qquad \mathbb{P}_{X_1, X_2, \ldots, X_n \overset{iid}{\sim} \mathcal{D}}[|\mathcal{A}(X_1, \ldots, X_n) - m(\mathcal{D})| \ge \alpha] \le \beta.$

*Proof.* Let $X = (X_1, \ldots, X_n)$ the $n$-tuple of i.i.d. samples from $\mathcal{D}$. Let us use a parameter $\gamma \in (0, 1)$ which we later choose a polynomial function of $\beta$. We consider the event

$$\mathcal{T}_\gamma := \{\exists X' \in \mathcal{H} \text{ s.t. } d_H(X, X') \le C' \log\frac{1}{\gamma}, m(X') = m(X)\},$$

where $C' = C'(C)$ is chosen to satisfy the conclusion of Lemma B.3. In particular it holds $C'(C) = \Theta(\frac{1}{C})$ as $C$ grows to infinity.

Notice that since $\alpha \leq r$ for some $n = O\left(\frac{\log\frac{1}{\gamma}}{L^2\alpha^2}\right)$ it holds $n \geq D\frac{\log\frac{1}{\gamma}}{L^2r^2}$ for the $D > 0$ defined in Lemma B.3. Furthermore, we can assume $C > 5$ is sufficiently large such that $4Ce\exp\left(-\frac{2C}{27}\right) < 1/2$. Hence, by applying Lemma B.3 we have

$$\mathbb{P}\left[|\mathcal{A}(X) - m(\mathcal{D})| \geq \alpha\right] \leq \mathbb{P}\left[|\mathcal{A}(X) - m(\mathcal{D})| \geq \alpha, X \in \mathcal{T}_\gamma\right] + \mathbb{P}\left[X \notin \mathcal{T}_\gamma\right]$$
$$\leq \mathbb{P}\left[|\mathcal{A}(X) - m(\mathcal{D})| \geq \alpha | X \in \mathcal{T}_\gamma\right] + \gamma. \tag{B.11}$$

Now conditioning on $X \in \mathcal{T}_\gamma$ we have that there exists an $X' \in \mathcal{H}$ with $d_H(X, X') \leq C'\log\frac{1}{\gamma}$ and $m(X') = m(X)$. Since the algorithm $\mathcal{A}$ is $\varepsilon$-differentially private by its definition, we have that $\mathcal{A}(X)$ and $\mathcal{A}(X')$ assigns to each output value the same probability mass up to a multiplicative factor of $e^{\varepsilon d_H(X,X')} \leq e^{\varepsilon C'\log\frac{1}{\gamma}} \leq e^{C'\log\frac{1}{\gamma}}$, where in the last inequality we use that $\varepsilon < 1$. Hence

$$\mathbb{P}\left[|\mathcal{A}(X) - m(\mathcal{D})| \geq \alpha | X \in \mathcal{T}_\gamma\right] \leq \mathbb{E}\left[e^{\varepsilon d_H(X,X')}\mathbb{P}\left[|\mathcal{A}(X') - m(\mathcal{D})| \geq \alpha | X \in \mathcal{T}_\gamma\right]\right]$$

$$\leq \frac{e^{C'\log\frac{1}{\gamma}}}{\mathbb{P}\left[X \in \mathcal{T}_\gamma\right]} \max_{X' \in \mathcal{H}, m(X')=m(X)} \mathbb{P}\left[|\mathcal{A}(X') - m(\mathcal{D})| \geq \alpha\right]$$

$$\leq \frac{e^{C'\log\frac{1}{\gamma}}}{1 - \gamma} \max_{X' \in \mathcal{H}, m(X')=m(X)} \mathbb{P}\left[|\mathcal{A}(X') - m(\mathcal{D})| \geq \alpha\right]$$

$$= \frac{e^{C'\log\frac{1}{\gamma}}}{1 - \gamma} \max_{X' \in \mathcal{H}, m(X')=m(X)} \mathbb{P}\left[|\hat{\mathcal{A}}(X') - m(\mathcal{D})| \geq \alpha\right], \tag{B.12}$$

$$\tag{B.13}$$

where in the last line we use that for all $X' \in \mathcal{H}$, it holds $\mathcal{A}(X') \overset{d}{=} \hat{\mathcal{A}}(X')$.

According to Lemma B.1 for some $n = O\left(C\frac{\log\frac{1}{\gamma}}{\varepsilon L\alpha} + \frac{\log\left(\frac{R}{\alpha}+1\right)}{\varepsilon Lr}\right)$ we can guarantee

$$\max_{X' \in H, m(X')=m(X)} \mathbb{P}\left[|\hat{\mathcal{A}}(X) - m(X)| \geq \frac{\alpha}{2}\right] \leq \gamma. \tag{B.14}$$

Using classical results we have that for some $n = O\left(\frac{\log(\frac{1}{\gamma})}{L^2\alpha^2}\right)$ it holds

$$\mathbb{P}[|m(X) - m(\mathcal{D})| \geq \frac{\alpha}{2}] \leq \gamma. \tag{B.15}$$

Combining the (B.14) and (B.15) we have for some $n = O\left(\frac{\log(\frac{1}{\gamma})}{L^2\alpha^2} + C\frac{\log\frac{1}{\gamma}}{\varepsilon L\alpha} + \frac{\log\left(\frac{R}{\alpha}+1\right)}{\varepsilon Lr}\right)$ it holds

$$\max_{X' \in H, m(X')=m(X)} \mathbb{P}\left[|\hat{\mathcal{A}}(X) - m(\mathcal{D})| \geq \alpha\right] \leq 2\gamma. \tag{B.16}$$

or, combining with (B.12),

$$\mathbb{P}\left[|\mathcal{A}(X) - m(\mathcal{D})| \geq \alpha | X \in \mathcal{T}_\gamma\right] \leq \frac{e^{C'\log\frac{1}{\gamma}}}{1 - \gamma}2\gamma.$$

or, combining with (B.11),

$$\mathbb{P}\left[|\mathcal{A}(X) - m(\mathcal{D})| \geq \alpha\right] \leq \frac{e^{C'\log\frac{1}{\gamma}}}{1 - \gamma}2\gamma + \gamma.$$

Since $C'(C) = \Theta(\frac{1}{C})$ we can assume $C > 0$ sufficiently large so that $C'(C) < \frac{1}{2}$. Hence for these values of $C$ some $n = O\left(\frac{\log(\frac{1}{\gamma})}{L^2\alpha^2} + C\frac{\log\frac{1}{\gamma}}{\varepsilon L\alpha} + \frac{\log\left(\frac{R}{\alpha}+1\right)}{\varepsilon Lr}\right)$ it holds

$$\mathbb{P}\left[|\mathcal{A}(X) - m(\mathcal{D})| \geq \alpha\right] \leq \frac{\sqrt{\gamma}}{1 - \gamma}2\gamma + \gamma.$$

Choosing $\gamma$ appropriately of the order $\gamma = \Theta\left(\beta^{\frac{2}{3}}\right)$ we conclude that

$$n = O\left(\frac{\log(\frac{1}{\beta})}{L^2\alpha^2} + C\frac{\log\frac{1}{\beta}}{\varepsilon L\alpha} + \frac{\log\left(\frac{R}{\alpha}+1\right)}{\varepsilon Lr}\right)$$

it holds

$$\mathbb{P}\left[|\mathcal{A}(X) - m(\mathcal{D})| \geq \alpha\right] \leq \beta.$$

The proof is complete.

$\square$

## C  Proof for Section 3.4: Lower Bounds

*Proof of Proposition 3.7.* We argue by contradiction and consider an algorithm $\mathcal{A}$ satisfying the negation of the statement of the proposition.

We first prove that

$$n = \Omega\left(\frac{r^2\log(\frac{1}{\beta})}{\alpha^2}\right) \tag{C.1}$$

and then

$$n = \Omega\left(\frac{\log\left(\frac{1}{\beta}\right)}{L^2\alpha^2}(1-2Lr)\right). \tag{C.2}$$

Notice that combined the lower bounds imply

$$n = \Omega\left(\frac{r^2\log(\frac{1}{\beta})}{\alpha^2} + \frac{\log\left(\frac{1}{\beta}\right)}{L^2\alpha^2}(1-2Lr)\right) = \Omega\left(\frac{\log(\frac{1}{\beta})}{\alpha^2}\frac{(1-Lr)^2}{L^2}\right) = \Omega\left(\frac{\log\left(\frac{1}{\beta}\right)}{L^2\alpha^2}\right),$$

where for the last equality we use $Lr \leq \frac{1}{2}$. The last displayed equation yields the desired contradiction.

To prove C.1 notice that all uniform distributions which are supported on an interval of width $2r$ inside $[-R-r, R+r]$ are admissible. Observe now that for a uniform distribution the mean and the median of it are identical. For this reason, $\mathcal{A}$ to satisfy the negation of our statement, it should learn the mean of these uniform distributions of width $r$ from $n$ samples with accuracy $\alpha$ with probability $1-\beta$. Standard learning theory implies that it should hold $n = \Omega\left(\frac{r^2\log(\frac{1}{\beta})}{\alpha^2}\right)$.

We now turn to C.2. Notice that if $Lr = \frac{1}{2}$ the lower bound on $n$ is trivial. Hence, to establish C.2, we focus on the case where $Lr < \frac{1}{2}$.

Recall the standard fact that learning the parameter $p$ of a Bernoulli random variable $\mathrm{Bernoulli}(p)$ at accuracy $\gamma > 0$ with probability $1-\beta$ requires $\Omega\left(\frac{\log\left(\frac{1}{\beta}\right)}{\gamma^2}\right)$ samples.

We know fix $p$. We construct the admissible distribution $\mathcal{D}$ which assigns probability mass $(1-2Lr)(1-p)$ at $-2r$, probability mass $(1-2Lr)p$ at $2r$ and with probability $2Lr$ samples from the uniform distribution on $[-r, r]$. It can be easily checked that $\mathcal{D}$ is admissible with the assumed parameters and median $p\left(\frac{1}{L}-2r\right)+r-\frac{1}{2L}$. In particular, learning the median at accuracy $\alpha$ is equivalent with learning the parameter $p$ at accuracy $\alpha\frac{2L}{1-2Lr}$ which from our assumption on $Lr < \frac{1}{2}$ is $\Theta\left((1-2Lr)^{-1}L\alpha\right)$. Furthermore, notice that for learning the parameter $p$ using the samples from $\mathcal{D}$ one needs to focus only the samples from which are equal to either $2r$ or $-2r$. Therefore the task of learning the median of $\mathcal{D}$ with $n$ samples at accuracy $\alpha$, reduces to learning the parameter $p$ of a $\mathrm{Bernoulli}(p)$ distribution with $N_1 = \mathrm{Binom}(n, 1-2Lr)$ samples at accuracy $\Theta\left((1-2Lr)^{-1}L\alpha\right)$.

Using the standard fact mentioned above and this equivalence, we have that conditional on the event, call it $\mathcal{E}_{\alpha,\beta}(\mathcal{D})$, that we can learn the median of $\mathcal{D}$ with $n$ samples at accuracy $\alpha$ it holds

$$N_1 = \Omega\left(\frac{\log\left(\frac{1}{\beta}\right)}{\alpha^2 L^2}(1 - 2Lr)^2\right).$$

Now recall that we assume that $\mathcal{A}$ can learn the median of $\mathcal{D}$ with $n$ samples at accuracy $\alpha$ holds with probability at least $1 - \beta$. Hence $\mathbb{P}\left(\mathcal{E}_{\alpha,\beta}(\mathcal{D})\right) \geq 1 - \beta$. Combined with the last observation of the paragraph above, we conclude

$$\mathbb{E}[N_1] \geq \mathbb{P}\left(\mathcal{E}_{\alpha,\beta}(\mathcal{D})\right)\mathbb{E}[N_1|\mathcal{E}_{\alpha,\beta}(\mathcal{D})] \geq (1 - \beta)\Omega\left(\frac{\log\left(\frac{1}{\beta}\right)}{\alpha^2 L^2}(1 - 2Lr)^2\right).$$

Using that $\mathbb{E}[N_1] = (1 - 2Lr)n$ and that $\beta \in (0, \frac{1}{2})$ we have

$$(1 - 2Lr)n = \Omega\left(\frac{\log\left(\frac{1}{\beta}\right)}{L^2\alpha^2}(1 - 2Lr)^2\right).$$

Using that $Lr < \frac{1}{2}$ we have $n = \Omega\left(\frac{\log\left(\frac{1}{\beta}\right)}{L^2\alpha^2}(1 - 2Lr)\right)$. The proof is complete.

$\square$

*Proof of Proposition 3.8.* We argue by contradiction and consider an algorithm $\mathcal{A}$ satisfying the negation of our statement. Since $R > 2\alpha$, for $\eta > 0$ sufficiently small it holds $2\alpha + \eta < R$. We consider a $\mathcal{D}_1$ which assigns mass $\frac{1}{2} - Lr$ to $-r - R$, mass $\frac{1}{2} - Lr$ to $r + R$ and with probability $2Lr$ samples from the uniform distribution on $[-r, r]$ and a $\mathcal{D}_2$ which assigns mass $\frac{1}{2} - Lr$ to $-r - R$, mass $\frac{1}{2} - Lr$ to $r + R$ and with probability $2Lr$ samples from the uniform distribution on $[-r + 2\alpha + \eta, r + 2\alpha + \eta]$. Note that they are both admissible with the desired parameters and it also holds $m(\mathcal{D}_1) = 0, m(\mathcal{D}_2) = 2\alpha + \eta$.

Using the accuracy guarantee of $\mathcal{A}$ and that the medians of the two distribution have distance strictly bigger than $2\alpha$, it necessarily holds

$$\mathbb{P}_{X_1,X_2,\ldots,X_n \stackrel{iid}{\sim} \mathcal{D}_1}[|\mathcal{A}(X) - m(\mathcal{D}_2)| \leq \alpha] \leq \mathbb{P}_{X_1,X_2,\ldots,X_n \stackrel{iid}{\sim} \mathcal{D}_1}[|\mathcal{A}(X) - m(\mathcal{D}_1)| \geq \alpha] \leq \beta. \tag{C.3}$$

The two distribution can be coupled in the following way: we first sample from $\mathcal{D}_1$. If it falls in $[-r + 2\alpha + \eta, r]$ we keep the sample from $\mathcal{D}_1$ as is, and translate it by $+2r$ if it falls into $[-r, -r + 2\alpha + \eta]$, to form a sample from $\mathcal{D}_2$. In particular, notice that by sampling two $n$-tuples $X, X' \in \mathbb{R}^n$ of samples in an i.i.d. sense from $\mathcal{D}_1$ and $\mathcal{D}_2$ respectively, under this coupling the $d_H(X, X')$ follows a $\mathrm{Binom}(n, (2\alpha + \eta)L)$. Hence using the definition of $\varepsilon$-differential privacy

$$\mathbb{E}\left[e^{-\varepsilon d_H(X,X')}\mathbb{P}_{X' \stackrel{iid}{\sim} \mathcal{D}_1}[|\mathcal{A}(X') - m(\mathcal{D}_2)| \leq \alpha]\right] \leq \mathbb{P}_{X_1,X_2,\ldots,X_n \stackrel{iid}{\sim} \mathcal{D}_1}[|\mathcal{A}(X) - m(\mathcal{D}_2)| \geq \alpha], \tag{C.4}$$

where the expectation is over the coupling mentioned above. Combining (C.3), (C.4) and the assumption on the accuracy performance of $\mathcal{A}$ we have

$$\mathbb{E}\left[e^{-\varepsilon d_H(X,X')}\right] \leq \frac{\beta}{1 - \beta}.$$

Using the moment generating function of the Binomial distribution we conclude

$$\left(1 - L(2\alpha + \eta)\left(1 - e^{-\varepsilon}\right)\right)^n \leq \frac{\beta}{1 - \beta}$$

which by basic asymptotics since $\beta < 1/2$ translates to $n = \Omega\left(\frac{\log\left(\frac{1}{\beta}\right)}{\varepsilon L(\alpha + \eta)}\right)$. Since $\eta > 0$ can be taken arbitrarily small, the proof of the proposition is complete.

$\square$

*Proof of Proposition 3.9.* We argue by contradiction and consider an algorithm $\mathcal{A}$ satisfying the negation of our statement.

We consider the partition of the interval $[-R, R]$ into $N = \Omega\left(\frac{R}{\alpha} + 1\right)$ consecutive intervals of width $3\alpha$ and let $m_i, i = 1, 2, \dots, N+1$ be the endpoints of these intervals.

Now consider $N$ admissible distributions $\mathcal{D}_i, i = 1, 2, \dots, N$ which for each $i$, assign mass $\frac{1}{2} - Lr$ at each of the points $-2(R + r)$ and $2(R + r)$ and with probability $2Lr$ it draws a sample from the uniform distribution on $[m_i - r, m_i + r]$.

By assumption for all $i = 1, 2, \dots, N$ it holds

$$\mathbb{P}_{X_1, X_2, \dots, X_n \overset{iid}{\sim} \mathcal{D}_i}[|\mathcal{A}(X_1, X_2, \dots, X_n) - m_i| \leq \alpha] \geq 1 - \beta. \tag{C.5}$$

Now since for all $i = 2, \dots, N+1 \ (i \neq 1)$ the distributions $\mathcal{D}_1, \mathcal{D}_i$ differ only on the interval that they are uniform on which they fall with probability $2Lr$, we can straightforwardly couple the $n$-tuples $X, X'$ sampled in an i.i.d. fashion from $\mathcal{D}_1, \mathcal{D}_i$ such that $d_H(X, X')$ follows a $\text{Binom}\,(n, 2Lr)$. Hence from the definition of $\varepsilon$-differential privacy it holds for all $i = 2, 3, \dots, N+1$

$$\mathbb{E}\left[e^{-\varepsilon d_H(X, X')}\mathbb{P}_{X' \overset{iid}{\sim} \mathcal{D}_i}[|\mathcal{A}(X') - m_i| \leq \alpha]\right] \leq \mathbb{P}_{X \overset{iid}{\sim} \mathcal{D}_1}[|\mathcal{A}(X) - m_i|, \leq \alpha] \tag{C.6}$$

where the expectation of the left hand side is under the aforementioned coupling. Now using (C.5) and the moment generating function of the Binomial distribution, it holds for all $i = 2, 3, \dots, N+1$

$$(1 - \beta)\left(1 - 2Lr\left(1 - e^{-\varepsilon}\right)\right)^n \leq \mathbb{P}_{X \overset{iid}{\sim} \mathcal{D}_1}[|\mathcal{A}(X) - m_i| \leq \alpha]$$

Now notice that the intervals $[m_i - \alpha, m_i + \alpha], i = 2, 3, \dots, N$ are disjoint and therefore

$$(1 - \beta)N\left(1 - 2Lr\left(1 - e^{-\varepsilon}\right)\right)^n \leq \mathbb{P}_{X \overset{iid}{\sim} \mathcal{D}_1}\left[\bigcup_{i=2}^{N+1}\{|\mathcal{A}(X) - m_i| \leq \alpha\}\right] \leq 1.$$

By standard asymptotics and as $Lr \leq \frac{1}{2}$, $\left(1 - 2Lr\left(1 - e^{-\varepsilon}\right)\right)^n = \Omega\left(e^{-Lr\varepsilon n}\right)$. Hence combining with the last displayed inequality, it holds

$$(1 - \beta)N = O\left(e^{Lr\varepsilon n}\right).$$

As $\beta < 1/2$ and $N = \Omega\left(\frac{R}{\alpha} + 1\right)$ we conclude

$$n = \Omega\left(\frac{\log\left(\frac{R}{\alpha} + 1\right)}{\varepsilon Lr}\right).$$

The proof is complete.

$\square$

# D   Insights about Sample Complexity

In this short section, we analyze the asymptotic behaviour of the optimal sample complexity for different ranges in the input parameters $(L, R, r, \beta)$ of the problem. Firstly, it is worth mentioning the intuition behind the three terms of $n_{\text{sc}}$:

$$n_{\text{sc}} = \underbrace{c_1 \frac{\log\left(\frac{1}{\beta}\right)}{L^2\alpha^2}}_{\text{Statistical Term } T_1} + \underbrace{c_2 \frac{\log\left(\frac{1}{\beta}\right)}{\varepsilon L\alpha}}_{\text{Privacy Local-Term } T_2} + \underbrace{c_3 \frac{\log\left(\frac{R}{\alpha} + 1\right)}{\varepsilon Lr}}_{\text{Privacy Global-Term } T_3}$$

Statistical Term $T_1$ is derived by the fact that the empirical median itself admits a sub-Gaussian error using Azuma's concentration bound. The other two $\varepsilon-$dependent privacy terms $T_2$ and $T_3$, of a smaller order in $1/\alpha$, are the price to pay in order to apply the truncated Laplace mechanism to the bounded-range median. Privacy Local-Term $T_2$ is derived by the fact that any private algorithm intrinsically can not differentiate easily local changes of order $d_H(X, X') = O(Lna)$. Privacy Global-Term $T_3$ is derived by the fact that any $\varepsilon$-differentially private algorithm needs to assign at least $\Omega(e^{-Lrn})$ probability mass in at least $R/\alpha + 1$ distinct intervals in $[-R, R]$.

More interestingly, there exist two critical values $\varepsilon_{\text{crit}}(\alpha, \beta, L, r, R), \alpha_{\text{crit}}(\beta, L, r, R)$ that suffice to determine which of the three aforementioned terms is the dominant one in the sample complexity.

1. (*"Privacy for free"*-regime) : If $\varepsilon > \varepsilon_{\text{crit}}(\alpha, \beta, L, r, R) = L\alpha \max\left\{1, \dfrac{\alpha \log(\frac{R}{a} + 1)}{\log(\frac{1}{\beta})r}\right\}$,

   then $n_{\text{crit}} = \Theta(T_1) = \Theta(\frac{1}{\alpha^2})$

2. (*"Local changes matter"*-regime) : If $\varepsilon < \varepsilon_{\text{crit}}$ and $\alpha > \alpha_{\text{crit}}(\beta, r, R) = \dfrac{\log(\frac{1}{\beta})r}{\log\left(\frac{R}{r\log(\frac{1}{\beta})}\right)}$,

   then $n_{\text{sc}} = \Theta(T_2) = \Theta(\frac{1}{\alpha\varepsilon})$

3. (*"Range of median matters"*-regime) : If $\varepsilon < \varepsilon_{\text{crit}}$ and $\alpha < \alpha_{\text{sc}}$, then $n_{\text{sc}} = \Theta(T_3) = \Theta(\frac{1}{\varepsilon})$

Figure 2: Contour plot of $n_{\text{sc}}$. The lighter colors indicate higher number of samples.

For fixed $(\beta, L, r, R)$ parameters, the behavior of the sample complexity is as follows: As long as $\varepsilon > \varepsilon_{\text{crit}}$ —our algorithm requires low privacy guarantees— the main bottleneck is the statistical task of computing an accurate approximate median of the distribution. Thus the dominant term is $T_1$. While our algorithm does not overstress the accuracy requirements and $\alpha > \alpha_{\text{crit}}$ and it becomes highly private and $\varepsilon < \varepsilon_{\text{crit}}$ the main bottleneck is the differential private task. The transition is continuous in the asymptotic area of $\varepsilon = \Theta(\frac{1}{\alpha^2})$. Finally when algorithm tries to be simultaneously extremely accurate and private the sample complexity is dominated by the combined term $T_2$.

# E   The Algorithm

In this section, we will present a polynomial-time algorithm which implements efficiently the aforementioned $\varepsilon$-differentially private estimator $\mathcal{A}$. The algorithm $\mathcal{A}$ has been proven to be rate-optimal in Theorem 3.6 which is the case the tuning parameter $C > 0$ is a sufficiently large constant. Here, we provide a sketch of our algorithm.

---

**Algorithm 1:** PRIVATEMEDIAN$(X = (X_1, \ldots, X_n))$

---
Sort $X$ s.t $X_1 \leq \ldots \leq X_n$.
$m(X) \leftarrow$ LEFTMEDIAN$(X)$
Let $\mathcal{I} \equiv [-R - 4Cr, R + 4Cr] \equiv [-B, B]$
Let
$$\begin{cases} \mathcal{D}_{\text{restricted}} \sim \exp\left(-\frac{\varepsilon}{4}\min\left\{\frac{Ln}{3C}|m(X) - \omega|, Lrn\right\}\right), \omega \in \mathcal{I} \\ \mathcal{D}_{\text{general}} \sim \exp\left(\inf_{X' \in \mathcal{H}}\left[\left(\frac{\varepsilon}{2}d_H(X, X')\right) - \frac{\varepsilon}{4}\min\left\{\frac{Ln}{3C}|m(X') - \omega|, Lrn\right\}\right]\right), \omega \in \mathcal{I} \end{cases}$$
**if** $X \in \mathcal{H}$ **then**
    $\lfloor$   $m \leftarrow$ Sample from $\mathcal{D}_{\text{restricted}}$ (Case (A))
**else**
    $\lfloor$   $m \leftarrow$ Sample from $\mathcal{D}_{\text{general}}$ (Case (B))
**return** $m$

---

It is easy to check that the running time of PRIVATEMEDIAN assuming a sorted data-set $X$ depends actually on the computational difficulty to sample from either $\mathcal{D}_{\text{restricted}}$ or $\mathcal{D}_{\text{general}}$, since the query $X \in \mathcal{H}$ can be answered with counting binary-searches in $O(n \log n)$. By inspection of the expressions of distributions $\mathcal{D}_{\text{restricted}}$ and $\mathcal{D}_{\text{general}}$, the main bottleneck of the worst-case analysis is the sampling process of $\mathcal{D}_{\text{general}}$, since it involves a challenging constrained optimization problem.

Recall that the typical set $\mathcal{H}$ as defined in 3.3, and in particular the PRIVATEMEDIAN algorithm, is a function of the parameter $C > 1$. The parameter $C > 1$ is treated as a constant in the main body of the present paper where we establish the optimality of the algorithm.

In Appendix E.1, we will show that even under the pessimistic model of worst-case analysis, our mechanism runs at most in $\text{poly}(n)$ time. The worst-case analysis below hold under arbitrary $C > 1$. On the other hand, in our average-case analysis presented in Appendix E.2 we assume that $C$ is scaling logarithmically with $n$. We show that this sacrifices the optimality of our sample complexity just up to a logarithmic factors and in that case our mechanism actually runs on expectation in $\tilde{O}(n)$ time.

*Note:For our worst-case and average-case analysis, we will assume that there is an oracle $\mathcal{O}$ such that we can always sample from Bernoulli, Uniform, Laplace, (Negative) Exponential Distribution and their truncated versions in an interval $I$ in $O(1)$ time. For more details, check appendix E.3*

## E.1   Worst-case Time Complexity of PRIVATEMEDIAN$(X = (X_1, \ldots, X_n))$

We start by presenting a simple random generator for $\mathcal{D}_{\text{restricted}}$ (Case (A)):

**Lemma E.1.** *For a given a date-set $X = (X_1, \cdots, X_n)$ and its median $m(X)$, there exists a $O(1)-$protocol that generates a sample from $\mathcal{D}_{\text{restricted}}$.*

*Proof.* First, notice that in the case of a data-set $X$ which belongs to the typical set $\mathcal{H}$, our estimator corresponds to sampling from the "flattened-Laplacian" continuous distribution of $\hat{\mathcal{A}}(X')$ as defined in Equation (3.4).

Therefore, its implementation would correspond to a simple two-phase protocol. In the first round we would flip an appropriately biased coin to decide between the two regions: the region where the estimator samples proportional to the Laplacian density and the region where the estimator samples proportional to the uniform distribution. In the second round we would simply apply conditional sampling either from a Laplacian or a uniform distribution, depending on the outcome of the coin flip. More specifically:

"The flattened"-Laplacian Mechanism: $\mathcal{D}_{\text{restricted}} \sim \exp\left(-\frac{\varepsilon}{4}\min\left\{\frac{Ln}{3C}\left|m(X)-\omega\right|, Lrn\right\}\right)$

---

**Algorithm 2:** Sample from $\mathcal{D}_{\text{restricted}}$

---

Let $I_{\text{left}}, I_{\text{center}}, I_{\text{right}}$ be the decomposition of $\mathcal{I} = [-B, B]$ as described in the above figure

// For extreme values of $C, r, m(X)$,

// either $I_{\text{left}}, I_{\text{center}}$ or $I_{\text{right}}$ may equal to $\varnothing$

Let $\begin{cases} p_{\text{center}} &= \displaystyle\int_{I_{\text{center}}} \exp\left(-\frac{\varepsilon}{4}\frac{Ln}{3C}\left|m(X)-\omega\right|\right)\mathrm{d}\omega \\[2mm] p_{\text{left}} &= \displaystyle\int_{I_{\text{left}}} \exp\left(-\frac{\varepsilon}{4}Lrn\right)\mathrm{d}\omega \\[2mm] p_{\text{right}} &= \displaystyle\int_{I_{\text{right}}} \exp\left(-\frac{\varepsilon}{4}Lrn\right)\mathrm{d}\omega \end{cases}$

Let $p = p_{\text{left}} + p_{\text{center}} + p_{\text{right}}$

Toss a trinary coin $c := \{\mathcal{L}, \mathcal{C}, \mathcal{R}\}$ with probability $(\frac{p_{\text{left}}}{p}, \frac{p_{\text{center}}}{p}, \frac{p_{\text{right}}}{p})$, correspondingly.

**if** $c$ *outputs* $\mathcal{L}$ **then**
  $\quad \lfloor \quad s \leftarrow$ Sample from UNIFORM$[I = I_{\text{left}}]$

**if** $c$ *outputs* $\mathcal{C}$ **then**
  $\quad \lfloor \quad s \leftarrow$ Sample from TRUNCATEDLAPLACE$[\mu = m(X), \sigma = \frac{12C}{Ln\varepsilon}, I = I_{\text{center}}]$

**if** $c$ *outputs* $\mathcal{R}$ **then**
  $\quad \lfloor \quad s \leftarrow$ Sample from UNIFORM$[I = I_{\text{right}}]$

**return** $s$

---

$\square$

We continue by presenting the construction of our sample generator for $\mathcal{D}_{\text{general}}$ (Case (B)). Firstly, let's recall the un-normalized term of $\mathcal{D}_{\text{general}}$, using the definition of $f_{\hat{\mathcal{A}}(X)}$ (Equation (3.6)):

$$\text{UNNORMALIZED}(X, \omega) = \exp\left(\inf_{X' \in \mathcal{H}}\left[\left(\frac{\varepsilon}{2}d_H(X, X')\right) - \frac{\varepsilon}{4}\min\left\{\frac{Ln}{3C}\left|m(X')-\omega\right|, Lrn\right\}\right]\right)$$

defined in $[-B, B] = [-R - 4Cr, R + 4Cr]$.

Our main technical contribution for this part is showing that the above constrained optimization problem can be solved in polynomial-time and then using it to obtain a polynomial-time sampler for the desired distribution. A key observation is that $\text{UNNORMALIZED}(X, \omega)$ depends only on the possible values of the median $m(X')$ and the distance $d_H(X, X')$, for any data-set $X' \in \mathcal{H}$.

To motivate our first technical lemma let's re-write the above optimization expression with an equivalent form

$$\text{UNNORMALIZED}(X, \omega) = \exp\left(\inf_{k \in [n]} \inf_{\substack{X' \in \mathcal{H} \\ m(X') = \xi \\ \xi \in [-R - r/2, R + r/2] \\ d_H(X, X') = k}} \left[\left(\frac{\varepsilon}{2}k\right) - \frac{\varepsilon}{4} \min\left\{\frac{Ln}{3C}|\xi - \omega|, Lrn\right\}\right]\right)$$

Therefore, in order to discretize the optimization space over $\mathcal{H}$, we use the following observation that we presented in Lemma 4.1.

**Lemma E.2** (Restated Lemma 4.1).
*Let* $\text{TYPICALHAMMING}(X, \xi) = \min_{X' \in \mathcal{H}, m(X')=\xi} d_H(X, X')$. *Then for any data-set* $X$, $\text{TYPICALHAMMING}(X, \cdot)$ *is piece-wise constant function in* $[-R - r/2, R + r/2]$ *with at most* $\text{poly}(n)$ *changes. Additionally, for any* $\xi$ *and data-set* $X$, $\text{TYPICALHAMMING}(X, \xi)$ *can be computed exactly using the* $\text{poly}(n)$ [4]-*time algorithm described in Algorithm 3.*

*Proof.* Our proof is divided into two parts:

1. (Claim E.3) We begin the proof by presenting a simple greedy algorithm, Algorithm 3, that outputs a data-set $X'$ such that it belongs to the typical set $\mathcal{H}$, and its median $m(X')$ equals to an input value $\xi$. From all the possible choices of $X'$ the greedy algorithm chooses the one that minimizes the Hamming distance from an input data-set $X = (X_1, X_2, \ldots, X_n)$.

2. (Claim E.4) Having established the correctness of the greedy method, we show that there exists a partition of $[-R - r/2, R + r/2]$ to a collection of $\text{poly}(n)$ disjoint subintervals such that in every subinterval, the output of the greedy algorithm remains constant.

**Claim E.3.** *The procedure Algorithm 3 for solving* $\text{TYPICALHAMMING}(X = (X_1, \ldots, X_n), \xi)$ *outputs a data-set* $X'$ *satisfying the conditions that* $m(X') = \xi$ *and* $X' \in \mathcal{H}$ *that minimizes the Hamming distance between* $X, X'$ *over all* $X' \in \mathcal{H}$, *i.e* $\min_{X' \in \mathcal{H}} \{d_H(X, X')\}$. *Moreover* $\text{TYPICALHAMMING}$ *runs in* $O(n^2)$ *time.*

*Proof.* Firstly, we will write down the list of the constraints that the above optimization algorithm needs to satisfy:

$$\begin{cases} \sum_{i \in [n]} \mathbf{1}\{\xi - X_i \in [0, +\infty]\} \geq \frac{n}{2} & \text{Median-Left Side} \\[2mm] \sum_{i \in [n]} \mathbf{1}\{X_i - \xi \in [0, +\infty]\} \geq \frac{n}{2} & \text{Median-Right Side} \\[2mm] \sum_{i \in [n]} \mathbf{1}\{\xi - X_i \in [0, \frac{C}{Ln}]\} \geq 2 & \kappa = 1\text{-Left Side} \\[2mm] \sum_{i \in [n]} \mathbf{1}\{X_i - \xi \in [0, \frac{C}{Ln}]\} \geq 2 & \kappa = 1\text{-Right Side} \\[2mm] \vdots \\[2mm] \sum_{i \in [n]} \mathbf{1}\{\xi - X_i \in [0, \frac{\lfloor \frac{Lnr}{2C} \rfloor C}{Ln}]\} \geq \lfloor \frac{Lnr}{2C} \rfloor + 1 & \kappa = \lfloor \frac{Lnr}{2C} \rfloor\text{-Left Side} \\[2mm] \sum_{i \in [n]} \mathbf{1}\{X_i - \xi \in [0, \frac{\lfloor \frac{Lnr}{2C} \rfloor C}{Ln}]\} \geq \lfloor \frac{Lnr}{2C} \rfloor + 1 & \kappa = \lfloor \frac{Lnr}{2C} \rfloor\text{-Right Side} \end{cases}$$

**Algorithm 3:** TYPICALHAMMING($X = (X_1, \ldots, X_n), \xi$)

---

**If** $\xi \notin [-R - r/2, R + r/2]$ **then return** Impossible

Create a copy $X'$ of data-set $X$ i.e $X'_i \leftarrow X_i \forall\ i \in [n]$ Sort $X'$ s.t $X'_1 \leq \ldots \leq X'_n$ and permutation $\pi : [n] \to [n]$ with $X'_{\pi(i)} = X_i$.

Part 1.`Rebalance the data-set s.t` $\xi$ `is the statistical median` **if**
$\left( \text{LEFTMEDIAN}(X') \neq \xi \right)$ **then**

    Let $X'_k \leq \xi \leq X'_{k+1}$.

    **for** $i \in \left[ \left| \lceil \frac{n}{2} \rceil - k \right| \right]$ **do**

       **if** $k < \lceil \frac{n}{2} \rceil$ **then**

          Set $X'_{n-i+1} \leftarrow \xi$

       **else**

          Set $X'_i \leftarrow \xi$

Part 2.`Rebalance the data-set to achieve concentration around` $\xi$

**for** $\kappa \in \{ \lfloor \frac{Lnr}{2C} \rfloor, \cdots, 1 \}$ **do**

    $\text{Right} \leftarrow \kappa + 1 - \sum_{i \in [n]} \mathbf{1}\{X'_i - \xi \in [0, \frac{\kappa C}{Ln}]\}$

    **while** *Right* $> 0$ **do**

       $X'_{\arg\max X'} \leftarrow \xi$

       $\text{Right} \leftarrow \kappa + 1 - \sum_{i \in [n]} \mathbf{1}\{X'_i - \xi \in [0, \frac{\kappa C}{Ln}]\}$

    $\text{Left} \leftarrow \kappa + 1 - \sum_{i \in [n]} \mathbf{1}\{\xi - X'_i \in [0, \frac{\kappa C}{Ln}]\}$

    **while** *Left* $> 0$ **do**

       $X'_{\arg\min X'} \leftarrow \xi$

       $\text{Left} \leftarrow \kappa + 1 - \sum_{i \in [n]} \mathbf{1}\{\xi - X'_i \in [0, \frac{\kappa C}{Ln}]\}$

For all $i = 1, \ldots, n$ set $X'_i \leftarrow X'_{\pi(i)}$

**return** $d_H(X, X')$

---

Our algorithm, as described in Algorithm 3, runs based on the following greedy choice:

$$\begin{cases} \textit{setting the new values to the central value } \xi \\ \textit{rebalancing always firstly the further elements from } \xi \end{cases}$$

It is easy to check using classical exchange arguments for both part (A) and part (B) that the algorithm outputs an optimal solution. We will show that given any other, different, optimal data-set $X_{\mathcal{O}}$ we can transform it to the solution of the above mechanism without increasing the Hamming Distance .

Indeed, let's denote the list of the modifications from the set $X$ to $X_{\mathcal{O}}$, i.e $\mathcal{M}_{\mathcal{O}} = \{X_{o_1} \to X'_{o_1}, \cdots, X_{o_k} \to X'_{o_k}\}$. Firstly we can mention that the greedy choice of *setting the new values to the central value* $\xi$ is always as good as any other optimal choice, since $\xi$ is by definition the median of $X_{\mathcal{O}}$. This holds because $\xi$ is the central value for any of the interval constraints. Thus we can transform $\mathcal{M}_{\mathcal{O}}$ to $\mathcal{M}'_{\mathcal{O}} = \{X_{o_1} \to \xi, \cdots, X_{o_k} \to \xi\}$.

Secondly, it is easy to check because of the nesting nature of the constraints around the median that the greedy choice of *rebalancing always firstly the further elements from* $\xi$ is again always as good as any other optimal choice. Indeed, let $\mathcal{T}$ be the list of modifications provided by our method, $\mathcal{M}_{\mathcal{T}} = \{X_{T_1} \to \xi, \cdots, X_{T_\ell} \to \xi\}$. Indeed, if $\mathcal{M}_{\mathcal{T}}$ and $\mathcal{M}_{\mathcal{O}}$ differ in one change we can always follow the greedy's choice since it will satisfy **at least** the same number of constraints than any other solution.

$$P_i \quad \underset{\xi}{\uparrow} \quad P_{i+1}$$

Thus without never worsening the optimal solution, we can eliminate any differences by changing inductively $X_{\mathcal{O}}$ to the solution of TYPICALHAMMING$(X = (X_1, \ldots, X_n), \xi)$. Finally the time complexity of the algorithm is $O(n \log n) + O(n)$ for part (A) and $O(Lnr) \times O(n) = O(n^2)$ for part (B). $\qquad \square$

**Claim E.4.** *For a given data-set $X = (X_1, \ldots, X_n)$, there is an ordered collection of $O(n^2)$ points $\mathbb{P} = \{-R - r/2 = P_0, P_1, \cdots, P_k, P_{k+1} = R + r/2\}$ such that for every open interval $I = (P_{i-1}, P_i)$, it holds that :*

$$\text{For any } \xi_1, \xi_2 \in I : \text{TYPICALHAMMING}(X, \xi_1) = \text{TYPICALHAMMING}(X, \xi_2).$$

*Proof.* We will start by describing the set $\mathbb{P}$. For each point of our data-set we create $2\lfloor \frac{Lnr}{2C} \rfloor + 1$ **anchor**-points. More precisely for the point $X_i$, we set as anchor points:

$$\text{anchors}(X_i) = \left\{ X_i \pm \kappa \times \frac{C}{Ln} \middle| \forall \kappa \in \{0, 1, \cdots, \lfloor \frac{Lnr}{2C} \rfloor\} \right\}$$

Let $\mathbb{P}$ be equal with the union of all the anchor points plus the limit points of the interval $[-R - r/2, R + r/2]$:

$$\mathbb{P} = \bigcup_{i \in [n]} \text{anchors}(X_i) \cup \{-R - r/2, R + r/2\}$$

It is easy to see that we need $O(n^2)$ time for its construction:

---

**Algorithm 4:** CONSTRUCTION $- \mathbb{P}(X = (X_1, \ldots, X_n))$

---

$\mathbb{P} \leftarrow \{-R - r/2, R + r/2\}$
**for** $i \in [n]$ **do**
$\quad \mathbb{P} \leftarrow X_i \cup \mathbb{P}$
$\quad$ **for** $\kappa \in \{1 \cdots \lfloor \frac{Lnr}{2C} \rfloor\}$ **do**
$\quad \quad \mathbb{P} \leftarrow (X_i + \kappa \times \frac{C}{Ln}) \cup \mathbb{P}$
$\quad \quad \mathbb{P} \leftarrow (X_i - \kappa \times \frac{C}{Ln}) \cup \mathbb{P}$
**return** $\mathbb{P}$

---

Finally, we will show that for any consecutive points of $\mathbb{P}$, let $p_i, p_{i+1}$, the algorithm TYPICALHAMMING$(X, \xi)$ outputs the same value for any $\xi \in (p_i, p_{i+1})$.

$$P_i \qquad \underset{\xi}{\uparrow} \underset{\xi'}{\uparrow} \quad P_{i+1}$$

Indeed, let's assume any $\xi \neq \xi'$ such that $\xi, \xi' \in (p_i, p_{i+1})$, where $p_i, p_{i+1} \in \mathbb{P}$. Since between $\xi, \xi'$ we assumed that there is no point of $\mathbb{P}$, it easy to check that

$$\mathbf{1}\{p_a \leq \xi \leq p_b\} = \mathbf{1}\{p_a \leq \xi' \leq p_b\} \quad \forall p_a, p_b \in \mathbb{P} \qquad \text{(Oblivious Property)}$$

By inspection of the algorithm, it follows that all the decision of the algorithm have been taken based on the queries of the form:

$$\left\{ \begin{array}{ll} Q_1(\xi) & : \sum_{i\in[n]} \mathbf{1}\{\xi \geq X_i\} \geq \frac{n}{2} \qquad\qquad Q_2(\xi) \qquad : \sum_{i\in[n]} \mathbf{1}\{X_i \geq \xi\} \geq \frac{n}{2} \\ Q_3(\kappa,\xi) & : \sum_{i\in[n]} \mathbf{1}\{X_i \leq \xi \leq X_i - \frac{\kappa C}{Ln}]\} > \kappa + 1 \quad Q_4(\kappa,\xi) \quad : \sum_{i\in[n]} \mathbf{1}\{\frac{\kappa C}{Ln} \leq X_i \leq \xi\} > \kappa + 1 \end{array} \right\}$$

More precisely the queries $Q_1(\xi), Q_2(\xi)$ are used in Part(A) and $Q_3(\xi), Q_4(\xi)$ in the Part (B). Additonally, it is easy to check that any query calculates sums of indicators of the form of eq. (Oblivious Property). Therefore for any $\xi, \xi'$ that belong to the open interval $(p_i, p_{i+1})$, it holds that:

$$Q_i(\xi) = \text{TRUE} \Leftrightarrow Q_i(\xi') = \text{TRUE} \quad \forall i \in \{1,2,3,4\}$$

which implies that the output of the optimal algorithm is the same and therefore $\text{TYPICALHAMMING}(X,\xi) = \text{TYPICALHAMMING}(X,\xi')$ ☐

From the above lemma, we showed that $\text{TYPICALHAMMING}(X,\xi)$ stays constant inside an open subcover of $[-R-r/2, R+r/2]$. To complete our proof for Lemma E.2, we just extend the definition also in the breakpoints $\mathbb{P}$, i.e

$$\text{TYPICALHAMMING}(X,\xi) = \left\{ \begin{array}{ll} \text{TYPICALHAMMING}(X, P_0) & \xi \in I_1 := [P_0, P_0] \equiv -R - r/2 \\ \text{TYPICALHAMMING}(X, \frac{P_0+P_1}{2}) & \xi \in I_2 := (P_0, P_1) \\ \text{TYPICALHAMMING}(X, P_1) & \xi \in I_3 := [P_1, P_1] \equiv P_1 \\ \text{TYPICALHAMMING}(X, \frac{P_1+P_2}{2}) & \xi \in I_4 := (P_1, P_2) \\ \text{TYPICALHAMMING}(X, P_2) & \xi \in I_5 := [P_2, P_2] \equiv P_2 \\ \vdots & \\ \text{TYPICALHAMMING}(X, P_{O(n^2)}) & \xi \in I_{O(n^2)} := [P_{O(n^2)}, P_{O(n^2)}] \equiv R + r/2 \end{array} \right.$$

$$\text{TYPICALHAMMING}(X,\xi) = \overset{4}{\downarrow}\ \overset{3}{\downarrow}\ \overset{2}{\downarrow}\ \overset{2}{\downarrow}\ \overset{1}{\downarrow}\ \overset{1}{\downarrow}\ \overset{2}{\downarrow}\ \overset{2}{\downarrow}\ \overset{2}{\downarrow}\ \overset{3}{\downarrow}\ \overset{3}{\downarrow}\ \overset{5}{\downarrow}\ \overset{6}{\downarrow}$$

Figure 3: *For illustrative purpose, we present a "possible" representative sketch of the decomposition of our interval $\mathcal{I} = [-B, B]$ based on $\text{TYPICALHAMMING}(X,\xi)$*

☐

**Remark E.5.** *One can establish more results on the structure of $\text{TYPICALHAMMING}$ than the one presented in Lemma E.2; for example $\text{TYPICALHAMMING}$ can be proven to be left or right semi-continuous depending on the position of the left empirical median $m(X)$, implying an additional structure on the intervals where it is of constant value. For example, for the simple case of $\mathcal{H} = \mathbb{Z}^n$ and $X = \{1,2,3,4,5\}$ $\text{TYPICALHAMMING}$ is constant in $[1,2)[2,3), [3,3], (3,4], (4,5]$. Nevertheless, in order to avoid the complexity of the description of the rules of the continuity in the more complex typical set $\mathcal{H}$ that we examine in our work, we just included in the proof of Lemma E.2 separately all the breakpoints as separated intervals where the $\text{TYPICALHAMMING}$ is trivially of constant value.*

An immediate consequence of Lemma E.2 is that the following definition is well-defined.

**Definition E.6.** *For any data-set $X$, there is a partition of $[-R - r/2, R + r/2]$ to a collection $\mathbb{I}$ of $\text{poly}(n)$ consecutive intervals, $\mathbb{I} = \{I_1, \cdots, I_{\text{poly}(n)}\}$ for which we denote*

$$\text{TYPICALHAMMING}(X, I) := \text{TYPICALHAMMING}(X, \xi),$$

*for arbitrary $\xi \in I$.*

We also define the following object.

**Definition E.7.** *For each* $k \in [n]$, *let us define* $\mathbb{I}_k$ *the union of the intervals* $I$ *where* TYPICALHAMMING$(X, I) = k$:

$$\mathbb{I}_k = \bigcup_{I \in \mathbb{I}, \text{TYPICALHAMMING}(X,I)=k} I$$

*Additionally, let us define the limit points* $\xi_{k,\inf}, \xi_{k,\sup}$ *such that* TYPICALHAMMING$(X, I) = k$:

$$\xi_{k,\inf} = \texttt{left-endpoint}(\mathbb{I}_k) = \inf_{\xi \in \mathbb{I}_k} \xi, \quad \xi_{k,\sup} = \texttt{right-endpoint}(\mathbb{I}_k) = \sup_{\xi \in \mathbb{I}_k} \xi$$

*Notice that for every* $k \in [n]$ *we can compute in* $O(n^4)$ *time* $\xi_{k,\inf}, \xi_{k,\sup}$ We are ready now to prove our main technical lemma:

**Lemma E.8** (Restated Lemma 4.2). *For any given data-set* $X$, *there is a partition of* $[-B, B]$ *to a collection* $\mathbb{J}$ *of* $\text{poly}(n)$ *consecutive intervals,* $\mathbb{J} = \{J_1, \cdots, J_{\text{poly}(n)}\}$ *such that :*

$$\text{For any } J_i \in \mathbb{J} \quad : \text{UNNORMALIZED}(X, \omega) = \exp\left(\alpha_i \omega + \beta_i\right) \quad \forall \omega \in J_i$$

*Moreover, we can compute exactly* $\mathbb{J}$ *and the constants* $(\alpha_i, \beta_i)$ *for every* $J_i \in \mathbb{J}$ *in* $O(\text{poly}(n))$ *time.*[5]

*Proof.* Using TYPICALHAMMING we can reduce the optimization problem for a given $\omega$ as:

$$\text{UNNORMALIZED}(X, \omega) = \exp\left( \min_{k \in [n]} \quad \inf_{\substack{X' \in \mathcal{H} \\ m(X') = \xi \\ \xi \in [-R - r/2, R + r/2] \\ d_H(X, X') = k}} \left[ \left(\frac{\varepsilon}{2}k\right) - \frac{\varepsilon}{4} \min\left\{ \frac{Ln}{3C} |\xi - \omega|, Lrn \right\} \right] \right)$$

$$= \exp\left( \min_{k \in [n]} \inf_{\substack{\text{TYPICALHAMMING}(X, \xi) = k \\ \xi \in [-R - r/2, R + r/2]}} \left[ \left(\frac{\varepsilon}{2}k\right) - \frac{\varepsilon}{4} \min\left\{ \frac{Ln}{3C} |\xi - \omega|, Lrn \right\} \right] \right)$$

$$\text{(By Definition E.6)} = \exp\left( \min_{k \in [n]} \inf_{\xi \in \mathbb{I}_k} \left[ \frac{\varepsilon}{2}k - \frac{\varepsilon}{4} \min\left\{ \frac{Ln}{3C} |\xi - \omega|, Lrn \right\} \right] \right)$$

$$\text{(By Definition E.7)} = \exp\left( \frac{\varepsilon}{2} \min_{k \in [n]} \underbrace{\left\{ k - \frac{1}{2} \cdot \frac{Ln}{3C} \min\left\{ \max_{p \in \{\xi_{k,\inf}, \xi_{k,\sup}\}} |p - \omega|, 3Cr \right\} \right\}}_{h_k(\omega)} \right) \Leftrightarrow$$

$$\text{UNNORMALIZED}(X, \omega) = \exp\left( \frac{\varepsilon}{2} \min_{k \in [n]} h_k(\omega) \right) \tag{E.1}$$

Let's try to understand the form of the function $h_k(\omega)$

By careful case study of $h_k(\omega)$ and by inspection of the above figure it is not difficult to check that every $h_k(\omega)$ can be described actually by at most 4 different clauses [6] with linear expressions [7]. Thus, without loss of generality, we can assume that every $h_k(\omega)$ is of the form:

$$\forall k \in [n] : h_k(\omega) = \begin{cases} \alpha_1^{[k]}\omega + \beta_1^{[k]} & \omega \in [\rho_1^{[k]}, \rho_2^{[k]}] \equiv [-B, \xi_{\sup,k} - 3Cr] \\[2ex] \alpha_2^{[k]}\omega + \beta_2^{[k]} & \omega \in [\rho_2^{[k]}, \rho_3^{[k]}] \equiv [\xi_{\sup,k} - 3Cr, \frac{\xi_{\inf,k}+\xi_{\sup,k}}{2}] \\[2ex] \alpha_3^{[k]}\omega + \beta_3^{[k]} & \omega \in [\rho_3^{[k]}, \rho_4^{[k]}] \equiv [\frac{\xi_{\inf,k}+\xi_{\sup,k}}{2}, \xi_{\inf,k} + 3Cr] \\[2ex] \alpha_4^{[k]}\omega + \beta_4^{[k]} & \omega \in [\rho_4^{[k]}, \rho_5^{[k]}] \equiv [\xi_{\inf,k} + 3Cr, B] \end{cases}$$

where $BP^{[k]} = \{\rho_1^{[k]}, \cdots, \rho_5^{[k]}\}$ are the breakpoints of $h_k$ in which the function changes its expression.

Having established the form of every $h_k(\omega)$, we are now ready to solve the optimization problem of computing $\min_{k \in [n]} h_k(\omega)$. It is easy to verify that the minimum of those "triangular pulses" would correspond to a piece-wise linear function. In other words, until now we have showed that, for any given data-set $X$ there is a partition of $[-B, B]$ to a collection $\mathbb{J}$ consecutive intervals, $\mathbb{J} = \{J_1, \cdots, J_{|\mathbb{J}|}\}$ such that :

$$\text{For any } J_i \in \mathbb{J} \quad : \min_{k \in [n]} h_k(\omega) = \alpha_i \omega + \beta_i \quad \forall \omega \in J_i$$

or equivalently using eq. (E.1):

$$\text{For any } J_i \in \mathbb{J} \quad : \textsc{UnNormalized}(X, \omega) = \exp(\alpha_i \omega + \beta_i) \quad \forall \omega \in J_i$$

To conclude the proof of our lemma it suffices to show that the size of the collection $\mathbb{J}$ is polynomial and to describe a $\text{poly}(n)$-time procedure that computes the piece-wise linear function $\min_{k \in [n]} h_k$.

Observe that to calculate $\min_{k \in [n]} h_k(\omega)$, it suffices to know the relative ordering of $h_1(\omega), \cdots, h_n(\omega)$ for each $\omega \in [-B, B]$. By continuity of $h_k$, the relative order of a pair $h_k, h_\ell$ can not change inside an interval $J$ where there is no solution of the equation $h_k(\omega) = h_\ell(\omega)$. Let us have $\Pi^{[k,\ell]} = \pi_1^{[k,\ell]}, \cdots, \pi_{c_{k,\ell}}^{[k,\ell]}$ be the ordered set of solutions of $h_k(\omega) = h_\ell(\omega)$ for every $k \neq \ell$. Let us denote denote also as $\Pi = \bigcup_{k \neq \ell} \Pi^{[k,\ell]} \cup \{-B, B\}$. Observe that in any interval $J$ of consecutive points of $\Pi$, the relative ordering of $h_k(\omega), h_\ell(\omega)$ for any $k, \ell \in [n]$ does not change inside $J$. Thus the minimizing function $h_\star$ does not change inside the interval $J$. We can further subdivide the interval $J$, if it is needed into sub-intervals based on its breakpoints where $h_\star$ is linear.

We can now construct the promised $\mathbb{J}$:

---
**Algorithm 5:** $\text{CONSTRUCTION} - \mathbb{J}(X = (X_1, \dots, X_n))$

---
Sort $\Phi = \bigcup_{k \neq \ell} \Pi^{[k,\ell]} \cup \{-B, B\} \cup \bigcup_k BP^{[k]}$

$//\Phi = \{\varphi_0 = -B \leq \pi_1 \leq \cdots \leq \varphi_m \leq \varphi_{m+1} = B\}$

$\mathbb{J} \leftarrow \varnothing$

**for** $i \in \{1, \cdots, m+1\}$ **do**
    Set $J_i = [\varphi_{i-1}, \varphi_i]$
    $\mathbb{J} \leftarrow \mathbb{J} \cup J_i$
**return** $\mathbb{J}$

---

- To evaluate the size of $\mathbb{J}$, it suffices to understand the size of $\Pi$ and $\bigcup_k BP^{[k]}$. On the one hand, the size of $\bigcup_k BP^{[k]}$ is at most $4n$, since we have at most 4 clauses in $h_k$. On the other hand, $\Pi$ includes all the solutions of $h_k(\omega) = h_\ell(\omega)$ for every $k \neq \ell$. This size is upper bound by $\binom{n}{2} \times \max_{k \neq \ell} \Pi^{[k,\ell]}$. Observe that to solve $h_k(\omega) = h_\ell(\omega)$, we need to solve at most 16 linear equations [8], each having at most one solution, since the pairs $(\alpha_j^k, \beta_j^k)$ are unique. Thus $\mathbb{J} = O(n^2) + O(n) = O(n^2)$.

- The time complexity of construction $\mathbb{J}$ is the sum of the elapsed time to compute all the $4n$ breakpoints $BP^{[k]}$, which can be done in $O(n^4)$ and compute the solutions of $O(n^2)$ linear equations and sorting their union $O(n^2 \log n)$. Thus, indeed we can construct $\mathbb{J}$ in $\text{poly}(n)$ time.

- To compute the constants $(\alpha_i, \beta_i) : \min_k h_k(\omega) = \alpha_i \omega + \beta_i \ \forall \omega \in J_i$, we can start from left to right. We compute the relative ordering of $h_1(-B), \cdots, h_n(-B)$. For each solution $h_k(\omega) = h_\ell(\omega)$, we update the relative ordering and for each breakpoint of $h_k$ we update its current linear form.

---
**Algorithm 6:** $\text{CONSTRUCTION} - (\mathbb{J}, \boldsymbol{\alpha}, \boldsymbol{\beta})(X = (X_1, \dots, X_n))$

---
Sort $\Phi = \bigcup_{k \neq \ell} \Pi^{[k,\ell]} \cup \{-B, B\} \cup \bigcup_k BP^{[k]}$

$//\Phi = \{\varphi_0 = -B \leq \pi_1 \leq \cdots \leq \varphi_m \leq \varphi_{m+1} = B\}$

Compute $h_1(-B), \cdots, h_n(-B)$

Keep a list of the current order $h_k$: $\Sigma = \{h_{i_1} \leq \cdots \leq h_{i_n}\}$

Keep a list of the current liner form of $h_k$: $T = \{h_1 : (\alpha_1^{[1]}, \beta_1^{[1]}) \cdots h_n : (\alpha_1^{[n]}, \beta_1^{[n]})\}$

**for** $i \in \{1, \cdots, m+1\}$ **do**
    Set $J_i = [\varphi_{i-1}, \varphi_i]$
    Set $(\alpha_i, \beta_i) \leftarrow T[\min h_k]$
    **if** $\varphi_i \in \Pi$ // `The next point is crossing point of two functions` **then**
        Update the relative order $\Sigma$
    **if** $\varphi_i = \rho_j^{[k]}$ // `The next point is some of the 4 breakpoints of` $h_k$ **then**
        Update $T[h_k] = (\alpha_j^{[k]}, \beta_j^{[k]})$
**return** $\{(J_i, \alpha_i, \beta_i)\}_m$

---

$\square$

**Lemma E.9.** *For a given a date-set $X = (X_1, \cdots, X_n)$ and its median $m(X)$, there exists a* $\text{poly}(n)-$*time protocol that generates a sample from* $\mathcal{D}_{general}$.

*Proof.* Indeed, the implementation of $\mathcal{D}_{\text{general}}$ implementation would correspond to a simple two-phase protocol. In the first round we would sample from a discrete distribution on a $\text{poly}(n)$-cardinality domain to decide among the regions $J_1, \cdots, J_{\text{poly}(n)}$. In the second round we would simply apply conditional sampling from the corresponding either exponential or uniform which will be truncated on the region which was the outcome of the first round. More specifically:

---

**Algorithm 7:** Sample from $\mathcal{D}_{\text{general}}$

---

Execute CONSTRUCTION $- (\mathbb{J}, \boldsymbol{\alpha}, \boldsymbol{\beta})(X = (X_1, \ldots, X_n))$
Let $\{(J_i, \alpha_i, \beta_i)\}_m$ be the $\text{poly}(n)$-decomposition of $\mathcal{I} = [-B, B]$ as described in lemma E.8
// For any $J_i$ : UNNORMALIZED$(X, \omega) = \exp\left(\alpha_i \omega + \beta_i\right)$ $\quad \forall \omega \in J_i$
**for** $i \in [m]$ **do**
$\qquad p_i = \displaystyle\int_{J_i} \exp\left(\alpha_i \omega + \beta_i\right) \mathrm{d}\omega$
Let $p = \sum_{i \in [m]} p_i$
Toss a $m$-nary coin $c := \{1, \cdots, m\}$ with probability $(\frac{p_1}{p}, \cdots, \frac{p_m}{p})$, correspondingly.
**if** $c$ *outputs* $\mathcal{C}$ **then**
$\qquad s \leftarrow$ Sample from TRUNCATEDEXPONENTIAL $[\alpha = \alpha_i, \beta = \beta_i, I = J_i]$
**return** $s$

---

The correctness of our estimator is derived by Lemma E.8.As we showed, the size of $J = \{(J_i, \alpha_i, \beta_i)\}_m$ is $\text{poly}(n)$ and we need $\text{poly}(n)$ time to compute $\{(J_i, \alpha_i, \beta_i)\}_m$. Our final step is to toss a $m$-nary coin which needs an extra $\Theta(m) = \text{poly}(n)$ time. $\qquad\square$

## E.2 Average-case Time Complexity of PRIVATEMEDIAN($X = (X_1, \ldots, X_n)$)

In this section we discuss about the average-case time complexity of the algorithm PRIVATEMEDIAN($X = (X_1, \ldots, X_n)$) which implements the algorithm $\mathcal{A}$ defined in (3.6). We consider running the algorithm for $C = C_0 \log n$ for some appropriate constant $C_0 > 0$. Note that this is different from the main body of this paper where it discusses the algorithm for a sufficiently large constant $C > 0$ and establishes its rate-optimality. We establish the following Corollary establishing that the algorithm remains rate-optimal up to logarithmic factors for this value of $C$.

**Corollary E.10.** *Suppose $C = C_0 \log n$ for some constant $C_0 > 0$. Suppose also $\varepsilon \in (0, 1)$, $\mathcal{D}$ is an admissible distribution and $\mathcal{A}$ is the $\varepsilon$-differentially private algorithm defined in (3.6) for this value of $C$. Then for any $\alpha \in (0, r)$ and $\beta \in (0, 1)$ for some*

$$n = \tilde{O}\left(\frac{\log\left(\frac{1}{\beta}\right)}{L^2 \alpha^2} + \frac{\log\left(\frac{1}{\beta}\right)}{\varepsilon L \alpha} + \frac{\log\left(\frac{R}{\alpha} + 1\right)}{\varepsilon L r}\right)$$

*it holds* $\quad \mathbb{P}_{X_1, X_2, \ldots, X_n \overset{iid}{\sim} \mathcal{D}}[|\mathcal{A}(X_1, \ldots, X_n) - m(\mathcal{D})| \geq \alpha] \leq \beta$.

The corollary follows directly by applying for this value of $C$ the Theorem 3.6 and using the basic asymptotic formula that for $A = \Omega(1)$, it holds $n/\log n = \Omega(A)$ if and only if $n = \Omega(A \log A)$. In particular, Corollary E.10 combined with the results of Section 3.4 allows us to conclude that the algorithm where $C$ scaling logarithmically with $n$ remains rate-optimal up to logarithmic factors.

We now show that for appropriate tuning of the parameter $C_0 > 0$ and under a weak bound on the sampling size $n$, satisfied for instance by the third term in the almost optimal rate of Corollary E.10, the algorithm runs in average-case in $\tilde{O}(n)$ time.

**Theorem E.11.** *There exists constants $C_0, D_0 > 0$ such that if $C = C_0 \log n$ and $T$ is the termination of time of* PRIVATEMEDIAN($X = (X_1, \ldots, X_n)$) *for this choice of $C > 0$ the following holds. If $n \geq D_0 \frac{1}{Lr} \log(\frac{1}{Lr})$, then*

$$\mathbb{E}[T] = O(n \log n).$$

*Proof.* We have by the law of total expectation,

$$\mathbb{E}[T] = \mathbb{P}[X \in \mathcal{H}]\,\mathbb{E}[T|X \in \mathcal{H}] + \mathbb{P}[X \notin \mathcal{H}]\,\mathbb{E}[T|X \notin \mathcal{H}]$$
$$\leq \mathbb{E}[T|X \in \mathcal{H}] + \mathbb{P}[X \notin \mathcal{H}]\,\mathbb{E}[T|X \notin \mathcal{H}]. \tag{E.2}$$

We now consider each term separately.

In the case where $X \in \mathcal{H}$ we need to check that $X \in \mathcal{H}$ to confirm the inclusion to the set $\mathcal{H}$ and then sample from the $\mathcal{D}_{\text{restricted}}$ distribution. For the first part, standard $O(n \log n)$ counting binary searches and locating the value of the median $m(X)$ suffices to certify that $X \in \mathcal{H}$. Furthermore, given Lemma E.1, since we have located the median $m(X)$, it takes $O(1)$-time to sample from $\mathcal{D}_{\text{restricted}}$. In conclusion

$$\mathbb{E}[T|X \in \mathcal{H}] = O(n \log n). \tag{E.3}$$

For the second part, since the algorithm runs in worst case polynomial-time there is a constant $M > 0$ such that

$$\mathbb{E}[T|X \notin \mathcal{H}] = O(n^M). \tag{E.4}$$

Now we focus on bounding $\mathbb{P}[X \notin \mathcal{H}]$. Notice that by identical reasoning as the derivation of inequality (14) of [BA19] in the proof of Lemma 3 in [BA19] we have for $t = r/2 \in [0, r]$

$$\mathbb{P}(|m(X) - m(\mathcal{D})| \geq r/2) \leq 2e^{-nL^2 r^2/8}.$$

Since $|m(\mathcal{D})| \leq R$ we conclude

$$\mathbb{P}(m(X) \notin [-R - r/2, R + r/2]) \leq 2e^{-nL^2 r^2/8}. \tag{E.5}$$

Furthermore, notice that since $C = C_0 \log n$ by assuming $n$ is bigger than a constant, Lemma B.2 can be applied with $T = 1$. Note that for $T = 1$, all counting constraints of $\mathcal{H}$ are considered and therefore, combined with (E.5) we have

$$\mathbb{P}[X \notin \mathcal{H}] \leq O\left(C_0 \log n \exp\left(-\frac{C_0 \log n}{8}\right) + e^{-\Theta\left(L^2 r^2 n\right)}\right).$$

Combining with (E.4) we have

$$\mathbb{P}\left[X \notin \mathcal{H}\right] \mathbb{E}\left[T | X \notin \mathcal{H}\right] \leq O\left(n^M C_0 \log n \exp\left(-\frac{C_0 \log n}{8}\right) + n^M e^{-\Theta\left(L^2 r^2 n\right)}\right).$$

Now using standard asymptotics there are $C_0, D_0 > 0$ sufficiently large constants dependent on $M$ for which if $n \geq D_0 \frac{1}{Lr} \log(\frac{1}{Lr})$, then

$$\mathbb{P}\left[X \notin \mathcal{H}\right] \mathbb{E}\left[T | X \notin \mathcal{H}\right] \leq O\left(1\right).$$

Combining with (E.3) and (E.2) we conclude

$$\mathbb{E}\left[T\right] \leq O\left(n \log n\right). \tag{E.6}$$

This completes the proof of the Theorem. $\qquad\qquad\square$

### E.3 Comments about Truncated-Sampling

In this last subsection, for the sake of completeness, we will present how we can simulate a random generator for Categorical, Bernoulli, Uniform, Laplace, (Negative) Exponential Distribution and their truncated versions in an interval $I$ in $O(1)$ time and $O(1)$ random queries of $\text{Unif}[0, 1]$ using the standard "inverse-CDF" sampling method.

The general inverse-CDF sampling method works as follows:

```
//Suppose that we want to simulate a distribution D
s.t its cdf F_D is invertible.
```
1. Generate a random number $u$ from the standard uniform distribution in the interval $[0, 1]$, e.g. from $U \sim \text{Unif}[0, 1]$.

2. Find the inverse of the desired CDF, e.g. $F_{\mathcal{D}}^{-1}(x)$.

3. Output $X = F_{\mathcal{D}}^{-1}(u)$.

Indeed, the computed random variable $X$ follows the distribution $\mathcal{D}$, since
$$\Pr(X \le x) = \Pr(F_{\mathcal{D}}^{-1}(U) \le x) = \Pr(U \le F_{\mathcal{D}}(x)) = F_{\mathcal{D}}(x).$$

#### E.3.1 UNIFORM

We start with an arbitrary uniform distribution in an interval $I = [a, b]$:

---
**Algorithm 8:** UNIFORM $[I = [a, b]]$

---
$u \leftarrow$ Sample from $U \sim \text{Unif}[0, 1]$
$s \leftarrow a + (b - a) \times u$
**return** $s$

---

#### E.3.2 $k-$nary coin

We continue with a $k-$nary coin, known as Categorical distribution, which is a discrete probability distribution that describes the possible results of a random variable that can take on one of $k$ possible categories, with the probability of each category separately specified.

---
**Algorithm 9:** $k-$nary coin $\{o_1, \cdots, o_k\}$ with probability $p_1, \cdots, p_k$

---
$u \leftarrow$ Sample from $U \sim \text{Unif}[0, 1]$
Compute CDF vector $(q_0, q_1, q_2, \cdots, q_{k-1}, q_k) = (0, p_1, p_1 + p_2, \cdots, \sum_{i=1}^{k-1} p_i, \sum_{i=1}^{k} p_i = 1)$
**for** $i \in [k]$ **do**
    **if** $q_{i-1} \le s < q_i$ **then**
        $s \leftarrow o_i$
**return** $s$

---

#### E.3.3 TRUNCATEDEXPONENTIAL

Our next probability distibution is the truncated (Positive/Negative) Exponential distribution
$\mathcal{D}_{\text{TruncExponential}} \sim \exp(\alpha\omega + \beta) \quad \omega \in I = [\texttt{left}, \texttt{right}]$.

- If $\alpha = 0$ then the actual distribution is the uniform.
- If $\alpha \ne 0$ we get the CDF of $\mathcal{D}_{\text{TruncExponential}}$ is

$$F_{\mathcal{D}_{\text{TruncExponential}}}(x|\alpha, \beta) = \frac{\exp(\alpha x + \beta) - \exp(\alpha \cdot \texttt{left} + \beta)}{\exp(\alpha \cdot \texttt{right} + \beta) - \exp(\alpha \cdot \texttt{left} + \beta)}.$$

Thus we get that:

$$F_{\mathcal{D}_{\text{TruncExponential}}}^{-1}(x|\alpha,\beta) = \frac{1}{\alpha} \ln \Big( \exp\left(\alpha \cdot \texttt{left} + \beta\right) + x \left(\exp\left(\alpha \cdot \texttt{right} + \beta\right) - \exp\left(\alpha \cdot \texttt{left} + \beta\right)\right) \Big) - \beta$$

---

**Algorithm 10:** TRUNCATEDEXPONENTIAL $[\alpha : \text{scale}, \beta : \text{shift}, I = [\texttt{left}, \texttt{right}]]$

---

$u \leftarrow$ Sample from UNIFORM $\big[I = [0,1] = [F_{\mathcal{D}_{\text{TruncExponential}}}(\texttt{left}), F_{\mathcal{D}_{\text{TruncExponential}}}(\texttt{right})]\big]$

$s \leftarrow F_{\mathcal{D}_{\text{TruncExponential}}}^{-1}(u)$

**return** $s$

---

### E.3.4 TRUNCATEDLAPLACE

Our last probability distibution is the truncated Laplace distribution $D_{\text{TruncLaplace}} \sim \frac{1}{2\sigma} \exp\left(-\frac{|\omega - \mu|}{\sigma}\right)$ $\omega \in I = [\texttt{left}, \texttt{right}]$. Here we firstly present the CDF and its inverse of the classical non-truncated version of Laplace distribution.

- $F_{\mathcal{D}_{\text{Laplace}}}(x|\mu,\sigma) = \frac{1}{2} + \frac{1}{2}\operatorname{sign}(x - \mu)\left(1 - \frac{1}{\sigma}\exp\left(-\frac{|\omega - \mu|}{\sigma}\right)\right)$

- $F_{\mathcal{D}_{\text{Laplace}}}^{-1}(x|\mu,\sigma) = \mu + \sigma\operatorname{sign}(x - \frac{1}{2})\ln\left(1 - 2|x - 0.5|\right)$

$$F_{\mathcal{D}_{\text{TruncLaplace}}}(x|\mu,\sigma) = \frac{F_{\mathcal{D}_{\text{Laplace}}}(x|\mu,\sigma) - F_{\mathcal{D}_{\text{Laplace}}}(\texttt{left}|\mu,\sigma)}{F_{\mathcal{D}_{\text{Laplace}}}(\texttt{right}|\mu,\sigma) - F_{\mathcal{D}_{\text{Laplace}}}(\texttt{left}|\mu,\sigma)}.$$

Thus we get that:

$$F_{\mathcal{D}_{\text{TruncLaplace}}}^{-1}(x|\mu,\sigma) = F_{\mathcal{D}_{\text{Laplace}}}^{-1}\left(F_{\mathcal{D}_{\text{Laplace}}}(\texttt{left}|\mu,\sigma) + x \times \left(F_{\mathcal{D}_{\text{Laplace}}}(\texttt{right}|\mu,\sigma) - F_{\mathcal{D}_{\text{Laplace}}}(\texttt{left}|\mu,\sigma)\right)|\mu,\sigma\right)$$

---

**Algorithm 11:** TRUNCATEDLAPLACE $[\mu : \text{location}, \sigma : \text{scale}, I = [\texttt{left}, \texttt{right}]]$

---

$u \leftarrow$ Sample from UNIFORM $\big[I = [0,1] = [F_{\mathcal{D}_{\text{TruncLaplace}}}(\texttt{left}), F_{\mathcal{D}_{\text{TruncLaplace}}}(\texttt{right})]\big]$

$s \leftarrow F_{\mathcal{D}_{\text{TruncLaplace}}}^{-1}(u)$

**return** $s$

---

## Footnotes

[4]For the interested reader, both the number of changes is at most $3n^2$ and the time complexity is $O(n^4)$, but we will keep for simplicity the general expression $\text{poly}(n)$ in our formal statement

[5]Again, for the interested reader, both the number of intervals is at most $7n^2$ and the exact time complexity is $O(n^4)$, but we will keep for simplicity the general expression $\text{poly}(n)$ in our formal statement

[6] The existence/size of the triangular pulse depends actually by the distance of $\xi_{\inf,k}, \xi_{\sup,k}$. Similarly, the existence/duration of the constant pulse depends on the relation of $\{\xi_{\inf,k} + 3Cr, \xi_{\sup,k} - 3Cr\}$ with $\{-B, B\}$.

[7] The constant pulse can be achieved by setting the corresponding $\alpha_i^{[k]}$ zero.

[8] In the general case for two piece-wise linear functions with 4 clauses, the maximum number of intersections Leveraging their specific "triangular-pulse" form, it is easy to verify that the maximum number is 4 and with an even more detailed case study the maximum number of intersections is actually 2, because the slopes of the "triangular-pulse" is the same for all $h_k$ functions.