[Reviews · NeurIPS 2020]

Review 1

Summary and Contributions: The paper studies the problem of private median estimation under a rather strong condition on the data generating distribution. The provide an algorithm/estimator which achieves optimal sample complexity in all problem parameters - the optimality established in a lower bound result in the paper. Furthermore, the paper gives a polynomial time algorithm to implement the estimator. --------------- Post author feedback comments --------------- I have read the author feedback, and keep my score unchanged. I encourage the authors to add/elaborate the discussion about the assumption provided in the feedback to the paper.

Strengths: 1. The authors study an important problem and resolve it almost completely. The paper is mathematically rigorous and makes strong theoretical contributions. 2. The paper is very well-written, and guides the reader with good intuition behind the techniques used. 3. Besides the main result, this gives the first instance of a "natural" problem where extension given by Extension Lemma (in previous work) can be computed in polynomial time time, as the authors remark. 4. Another good thing is that the appendix is sufficiently polished and they cover a lot of background to make the paper self-contained - like they provide a proof of extension lemma adapted to their setting.

Weaknesses: 1. One question that I had was why this peculiar assumption - it seems strong, and perhaps something like this is necessary, but is it common in the statistical estimation literature (for median estimation say)? 2. Adaptivity - It seems like the algorithm requires knowledge of all the problem parameters, so is there a way to adapt to "easier" settings? 3. Another thing I found missing was discussion of techniques for private median estimation in the literature. I am not very familiar with it, so if it were provided, it would help to judge if/how much the authors leverage techniques already developed in the context of private median estimation.

Correctness: I skimmed through some of the proofs, and they appeared correct to me. Furthermore, the high-level ideas behind the proofs look sound.

Clarity: Yes.

Relation to Prior Work: Mostly, but would be good to include how much the techniques leverage previous work (see weakness 3).

Reproducibility: Yes

Additional Feedback: Some typos in proofs (small details): (a). In the proof of Proposition 3.7, the distribution $D$ does not seem to have total measure 1 -possibly a typo or my misunderstanding? Moreover, the median reported seems off by a factor of 4 in some terms. (b). In proof of proposition 3.8, I did not get why the median of $D_1$ is 0 and that of $D_2$ is $2\alpha+\eta$? My calculation, in case I am doing something wrong: for $D_1$, the total measure to the left of 0 is $1-Lr/2 + (Lr/2)(r)/2r = 1 - Lr/4$, and not half. (c). Typo in line 752: should me $m_i+r$.


Review 2

Summary and Contributions: This paper studies the basic problem of median estimation with differential privacy. Under fairly minimal distributional assumptions, namely that the data comes from a distribution with probability density bounded away from zero in a neighborhood of its median, the paper gives an algorithm for approximating the median under pure (eps, 0)-differential privacy. The algorithm is built using the framework of Lipschitz extensions, and in particular, invokes a general "extension theorem" of Borgs et al. With additional work, the authors are able to show that the resulting algorithm can be implemented in polynomial time. The sample complexity guarantee of the algorithm improves on the state-of-the-art in this setting; moreover, it achieves pure rather than approximate DP. The paper then gives lower bounds shows that the sample complexity guarantee of the new algorithm is tight.

Strengths: This is a complete theory paper: It addresses a fundamental problem, designs and analyzes a new algorithm, and proves the algorithm's optimality.

Weaknesses: It would be helpful to discuss the extent to which the definition of "admissible" is necessary for the results to hold. In particular, the results are robust to zeroing out the density at singletons near the median. Could a more natural condition just be that the distribution places sufficient mass on every small ball around the median?

Correctness: I did not check proofs in detail, but the claims and techniques seem correct.

Clarity: The main ideas of the paper are presented very clearly. The technical arguments in the appendix, however, quickly jump into calculations and could benefit from more informal discussion.

Relation to Prior Work: Prior work is discussed adequately. Some comparison could also be made to algorithms for median/mean estimation via smooth sensitivity: http://www.cse.psu.edu/~ads22/pubs/NRS07/NRS07-full-draft-v1.pdf and https://arxiv.org/abs/1906.02830.

Reproducibility: Yes

Additional Feedback: - The definition of (eps, delta)-DP in line 465 is incorrect when the Hamming distance between X and X' is greater than 1 - Line 125: extraneous "estimation of the" - Line 155: in on -> in EDIT: I thank the authors for their feedback and for the planned enhancements to the presentation of the work.


Review 3

Summary and Contributions: This paper gives the first differentially private polynomial-time sample-optimal algorithm for median estimation. (It also gives the lower bounds to prove the "sample-optimal" part.)

Strengths: This paper gives optimal results for a fundamental problem of broad interest.

Weaknesses: The basic techniques used in this paper, the Laplace mechanism and the extension lemma, are not particularly novel.

Correctness: The technical outline seems correct. The authors make a non-technical claim I am not sure about: is this really the "first instance of a natural problem" for which the [BCSZ18a] lemma gives an extension that can be computed in polynomial time, as the authors claim at the bottom of section 1.2? They say this "affirmatively" answers a question posed in that paper, but [BCSZ18a] asks "under which conditions such an extension can become computationally efficient" rather than a yes/no question.

Clarity: For the most part the exposition is good. Section 5 has a number of grammar errors. Missing a word after "private" on line 163. Eq. 3.4 has a q instead of an omega. "flattened" is misspelled in several different ways in various places. "assigns" is repeated on line 245

Relation to Prior Work: Yes

Reproducibility: Yes

Additional Feedback: I thank the authors for their feedback that clarified the novelty of their contribution.


Review 4

Summary and Contributions: Authors provide a polynomial-time differentially private algorithm for estimation of the median for a finite number of samples, assuming a positive density at a small neighborhood around the median of the considered distribution. Compared to previous work: 1) this is not required the distribution has bounded values 2) this is not required the distribution has finite moments 3) the median is learned under pure eps-Differential Privacy (vs (eps, delta)-DP), so it consists in stronger privacy guarantees. Proofs are based on a 2-step framework: 1) Define a private algorithm on a restricted class of instances that are "typical" for the distribution, based on a variant of the Laplace mechanism. 2) Extend a) the obtained estimator to all instances and b) the privacy guarantees based on an existing method for Lipschitz extensions

Strengths: There is systematic theoretical grounding of the claims followed by an interpretation in words of the results. Steps of the general proofs are well described (cf. summary and contributions). In general the paper is well written and very complete from the theoretic point of view. Authors are bringing novel results in a domain of ML that is gaining wide interest of the community: differential privacy in ML. So no doubt on the relevance of this work for the NeurIPS conference.

Weaknesses: Even if the contributions of the paper are theoretic, at least a small empirical evaluation can be expected. It would be nice to see, for instance: 1) how the bounds are behaving when playing with the parameters 2) what is the quality of the median estimation (compared to exact non-private value) in function of the different parameters (in particular depending on the number of samples, type of distribution, epsilon, etc.)

Correctness: It seems yes for the claims: they are proven, if not in the the main part of the part, in the supplementary material. Given the time for reviewing, I could not read all of them though.

Clarity: Yes, the paper is well written as the results are introduced in a very pedagogical way. Moreover each theoretical result is interpreted in words to ensure the reader follows the reasoning. Extensive supplementary material (that I did not entirely read though, to be honest) accompanies the paper to make it self-contained. Broader impact has been thoroughly addressed.

Relation to Prior Work: Yes for previous work on median estimation with privacy constraints and theoretic material used for the proofs. In particular, authors refer to Avella-Medina and Brunel for previous work on median estimation under arbitrary distributions that satisfy a mild concentration condition but whose median lies in a bounded range. Moreover, while this work was under the (eps, delta)-differential privacy setting, here authors provide stronger privacy guarantees with delta = 0.

Reproducibility: Yes

Additional Feedback: Minor details: It is convenient to add the link of the citation in the alias so that by clicking on it, the reader can easily reach the bibliography at the end. Some typos: -r144: extra dot -row 162: "Our results follows" -row 184: "flatenned" -row 245: "assigns assigns" -row 262: "depending in" -row 284: "consists a concatenation" -row 319: "spectrum ," -row 319: "consists the" === AFTER AUTHOR FEEDBACK === Authors successfully addressed my concerns regarding the lack of some experiments illustrating the theoretical results. I trust them to add some in the revised version of the paper. Hence I upgrade my note to an "accept".

[Author Response · NeurIPS 2020]

We wholeheartedly thank the reviewers for the positive and encouraging feedback, as well as for the insightful comments to improve our submitted work. We first also thank all the reviewers for finding multiple minor typos, which we commit on correcting in the revised version of our work. We now continue by addressing one comment appearing in multiple reviews (Reviewers #1, #2) and then proceed with addressing the additional comments of each reviewer separately.

*(Admissibility Assumption)* We thank the reviewers for asking more information regarding the admissibility assumption for our distribution. The assumption is meant to quantify the following intuitive statistical fact: to recover the median of a distribution from i.i.d. samples, the distribution needs to assign positive mass at every sufficiently small neighborhood around the median (and furthermore the less mass it assigns the harder it is to recover). An instance of this fact appears in a standard result in the theory of statistics (using e.g. the delta method) saying that the empirical median of i.i.d. random variables is only asymptotically normal when the data have a positive density at the true median, call if $f(m) > 0$ (the asymptotic variance of the empirical median becomes $4f(m)^{-2}$.) Note that $f(m) > 0$ indeed implies a positive mass in every sufficiently small neighborhood around the median. We adopted the specific quantification of the admissibility assumption directly from the earlier work of [BA19], [BA20] on private median estimation, so that we are in line with previous work. Yet, as suggested by Reviewer #2, indeed all our results can be generalized in a straightforward manner to the case where the distribution does not necessarily admit a density around the median in $[m(D) - r, m(D) + r]$ but still satisfies $|F(t) - F(s)| \geq L|t - s|$ for all $t, s \in [m(D) - r, m(D) + r]$, where $F$ is the CDF of the distribution. We consider this an interesting remark and commit on expanding on the above discussion at the revised version of our work.

**Reviewer #1**

*(Adaptivity to unknown problem parameters)* We thank the reviewer for asking whether one can design an optimal private estimator who is adaptive to the parameters, that is it remains optimal without knowledge of the values of $L, R, r$. While our methods are not tailored to work in this setting, we consider this question of high importance and a very interesting direction of future work.

*(Prior methods for private median estimation)* To the best of our knowledge, we are the first to apply Lipschitz extension techniques, such as the "Extension Lemma", and the "truncated Laplacian" ideas in the context of private median estimation.

*(Propositions 3.7 and 3.8)* We thank the reviewer for pointing out to us two important typos in the proofs of Propositions 3.7. and 3.8. We commit on correcting them in the revised version of this work.

**Reviewer #2**

*(Smooth sensitivity prior work)* We thank the reviewer for pointing us to the interesting use of smooth sensitivity in the prior work of private median estimation. We commit to adding a discussion in our literature review and the proposed citations.

**Reviewer #3**

*(Novelty)* While we agree with the reviewer that using Lipschitz extension techniques and Laplace noise are among the most frequently used techniques in differential privacy, the specific way used in this work have only been used recently in a similar manner for learning Erdos-Renyi models in [BCSZ18b]. Importantly, in our submitted work we build and expand on the methods of [BCSZ18b], since we not only we appropriately use the Lipschitz extension technique - "the Extension Lemma"- developed in [BCSZ18b], but we show how the extension can be made in polynomial time in our context.

*(The Extension Lemma- "non-technical claim")* The reviewer is absolutely correct that the question in [BCSZ18a] is posed in terms of conditions under which the extension can be made in polynomial time. We commit on appropriately adjusting the wording in our paper for the revised version of our papers. That being said, while the authors of [BCSZ18a] cite in their conclusion multiple papers performing Lipschitz extensions in polynomial-time in prior work, none of these papers use the (admittedly powerful) extension method which is used to prove the Extension Lemma in [BCSZ18a]. To the best of our knowledge we are the first who show that this extension method can be made in polynomial-time in our context.

**Reviewer #4**

*(Citations)* We agree to add the link of the citation in the alias so that by clicking on it, the reader can easily reach the bibliography at the end.

*(Empirical Evaluation- parameter dependence)* We agree with the reviewer that some discussion analyzing the dependence of the various parameters on the optimal rate would help the reader to build intuition. We commit on adding some informal discussion on that, coupled with appropriate empirical evaluations, in the revised version of our work.

[Meta-Review · NeurIPS 2020]

This paper addresses a basic problem in privacy-preserving data analysis, namely, the problem of differentially private mean estimation. Under minimal distributional assumptions, the paper provides tight upper and lower bounds on the sample complexity of this problem. Moreover, the constructed algorithm runs in polynomial time. These are strong theoretical results for a fundamental problem of broad interest.